# Preference-Enhanced Reinforcement Learning for Pluralistic Image Inpainting

**Peng Zhou** [1]   **Muqi Huang** [2]   **Tianshuo Qu** [1]   **JingYang Wang** [1]   **Kun Zhou** [2]   **Chuan Li** [2]   **Feng Shi** [2]   **Shi Chen** [3]   **Yun Xiong** [1]

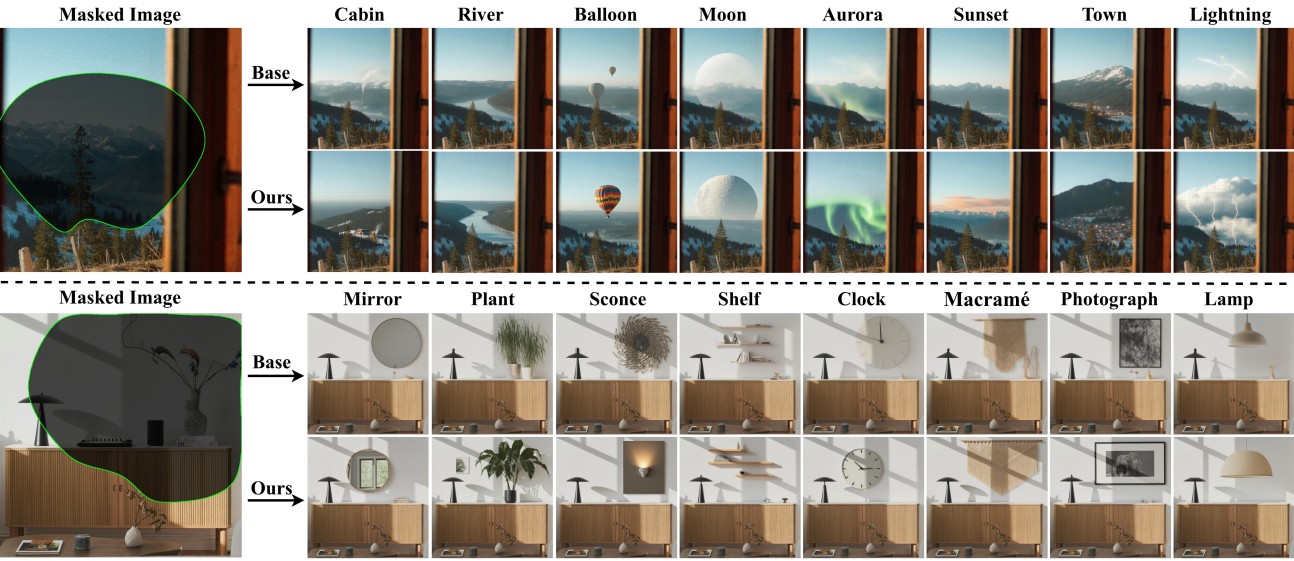

*Figure 1.* **Visual comparison of text-guided inpainting.** We compare FLUX1.fill (Base) and our method using diverse prompts for the masked region (labeled above each column; e.g., Cabin, River). Our method achieves superior visual aesthetics compared to the baseline.

## Abstract

Existing image inpainting frameworks rely on strictly supervised training paradigms, often suffering from an over-reliance on ground-truth reconstruction, which leads to conservative outputs with misaligned creativity and limited diversity. To this end, we propose the first framework to explore Group Relative Policy Optimization (GRPO) and Direct Preference Optimization (DPO) for text-guided image inpainting, formulating an efficient online reinforcement learning pipeline that enables flexible, human-aligned aesthetic control via a preference scoring model. Crucially, by decoupling the rigid one-to-one correspondence between text prompts and masked images, our method enables the model to explore diverse, controllable, and high-quality solutions beyond a single target. Furthermore, to balance semantic consistency with physical naturalness at mask boundaries, we introduce a scale-aware dynamic reward mechanism that adaptively emphasizes boundary gradient coherence for small occlusions while prioritizing visual aesthetics in large-scale generation. Extensive experiments demonstrate that our approach consistently produces higher-quality results across different backbone architectures such as Stable Diffusion and FLUX, significantly enhancing the generative capacity of base models.

[1]Shanghai Key Lab of Data Science, College of Computer Science and Artificial Intelligence, Fudan University, Shanghai, China [2]Rajax Network Technology (Taobao Shangou of Alibaba), Alibaba Group, Shanghai, China [3]Faculty of Science and Technology, University of Macau, Macau, China. Correspondence to: Muqi Huang <huangmuqi.hmq@alibaba-inc.com>, Yun Xiong <yunx@fudan.edu.cn>.

*Proceedings of the $43^{rd}$ International Conference on Machine Learning*, Seoul, South Korea. PMLR 306, 2026. Copyright 2026 by the author(s).

## 1. Introduction

With the emergence of diffusion models (DMs) (Rombach et al., 2022; Yang et al., 2024) and flow-matching Transformers (Peebles & Xie, 2023; Lipman et al., 2023; Esser et al., 2024), the field of generative computer vision has undergone

a paradigm shift. Among them, tasks such as Text-to-Image (T2I) (Ramesh et al., 2022; Podell et al., 2024) and Image-to-Image (I2I) (Zhang et al., 2023b; Brooks et al., 2023) have particularly benefited from these advances, achieving remarkable progress in both quality and controllability. Image inpainting, equally a mainstream application-oriented image generation task, also stands at the frontier of practical image editing (Xie et al., 2023). Unlike T2I, inpainting requires a delicate balance between being "faithful to user instructions" (Nichol et al., 2022) and maintaining "seamless consistency with the unmasked background" (Lugmayr et al., 2022; Avrahami et al., 2022).

Current mainstream inpainting models primarily rely on a ground-truth reconstruction supervised training paradigm (Kawar et al., 2023; Sheynin et al., 2024; Ju et al., 2024a; Manukyan et al., 2025). They use text describing the masked region as a condition and minimize reconstruction errors, such as Mean Squared Error (MSE), to drive the model to recover the ground-truth signal. This paradigm strictly anchors the optimization objective to maximizing the likelihood of a single Ground Truth (GT). This reveals a fundamental flaw when facing creative editing tasks: when user instructions allow for multiple reasonable outcomes (a one-to-many mapping) (Zhang et al., 2023a), forcing the model to fit a single GT leads to an over-reliance on ground-truth reconstruction, resulting in conservative outputs with limited diversity (Brooks et al., 2023; Wallace et al., 2024). As illustrated in Figure 1, Flux.1 Fill, currently the largest open-source inpainting model, falls into the trap of "probability averaging" when handling instructions requiring high semantic diversity. The model suffers severely from the "over-smoothing" problem. Consequently, generated images tend to be conservative and lack diversity, failing to meet aesthetic expectations.

To address the over-reliance on a single GT in conventional supervised training, we innovatively propose an online reinforcement learning (RL) framework for image inpainting that unlocks the generation potential of the model through policy alignment. Unlike offline fine-tuning with static datasets, our method treats inpainting as a sequential decision process: the model generates "online data" via iterative sampling and updates its policy using real-time feedback from a reward model. This architecture not only eliminates the strict constraint of precisely paired image-text datasets, but also enables continuous exploration of diverse visual-semantic spaces, facilitating more flexible content editing under open-domain conditions. Moreover, this training paradigm fully leverages the rich prior knowledge available in the era of large models, driving the model toward human aesthetic preferences without requiring additional collection of annotated data. We implement both Group Relative Policy Optimization (GRPO) (Shao et al., 2024; Liu et al., 2025) and Direct Preference Optimization (DPO) (Rafailov

et al., 2023; Wallace et al., 2024). Due to GRPO's superior stability and generalization, we use it as the primary optimizer while reporting both for reproducibility. However, applying RL suffers from low sensitivity to local changes in reward signals, as masked regions always occupy only a small portion of the image. Global reward models are thus dominated by the background, leading to negligible score differences between outputs, as shown in Figure 3. Moreover, general aesthetic metrics often fail to capture inpainting-specific "stitching traces". To address this, we design a dynamic reward mechanism: we introduce a gradient discontinuity penalty to quantify and eliminate edge artifacts; simultaneously, we employ an adaptive weight adjustment mechanism based on the mask ratio to set priorities for different editing scales—strictly adhering to physical boundaries for small occlusions, while emphasizing visual aesthetics for large-scale repainting.

In summary, our contributions[1] are as follows:

- We propose the first online alignment inpainting framework based on GRPO and DPO. Our method breaks the supervised training paradigm, enabling high-quality generation and human alignment through a lightweight reward feedback mechanism.

- We decouple strict text-image pairing constraint, allowing training on unpaired data. This design significantly expands data diversity and improves generalization in open-domain scenarios.

- We introduce a dynamic reward mechanism specifically for inpainting. Balances semantic consistency and boundary naturalness with negligible overhead, greatly enhancing mask-region coherence.

- We conduct cross-architecture validation, showing consistent performance improvements on both DiT and U-Net backbones, significantly enhancing editing capability under complex instructions.

## 2. Related Work

### 2.1. Image Inpainting Architectures

**Adapter-based Modular Approaches.** Early research augmented pre-trained T2I models with external inpainting modules. ControlNet-Inpainting (Zhang et al., 2023b) processed mask conditions using a trainable encoder copy. PowerPaint v1 (Zhuang et al., 2024) adopted this architecture for diverse editing tasks. ControlNet Union and ControlNet++ further advanced this direction with multi-task frameworks. They used conditional routing mechanisms to decou-

---

[1]Code is available at `https://drive.google.com/file/d/1MpsXje8K8MJJ8e2JS6fJ8GFJ1r5yh9m_/view?usp=sharing`.

ple multiple control signals (Zhao et al., 2023). To improve pixel fidelity, BrushNet (Ju et al., 2024b) proposed a dual-branch strategy. This method separates masked features from generated ones to ensure context consistency. Power-Paint v2 (Zhuang et al., 2024) later incorporated this design to enhance training stability. Additionally, TurboFill (Xie et al., 2025) optimized the architecture for real-time use through adversarial training. However, these modular approaches require many additional parameters. This increases inference latency and limits practical application.

**Native Inpainting Models.** Models like FLUX.1 Fill (Black Forest Labs, 2024) and Stable Diffusion Inpainting (Rombach et al., 2022) have shifted the focus toward native training paradigms. These approaches integrate mask channels directly into the base architecture. FLUX.1 Fill uses the Flow Matching architecture to achieve superior consistency without external adapters. OmniGen (Xiao et al., 2025) explores a unified paradigm by interleaving image and text tokens for instruction-driven editing. Despite these advancements, such models rely primarily on GT supervision. They often lack alignment with complex human preferences. This limits their generalization in open domains. Consequently, a post-training alignment phase is necessary.

## 2.2. Preference Alignment for Image Generation

**Offline Preference Alignment.** Many T2I studies use offline alignment to avoid the high costs of online sampling. Following Diffusion-DPO (Wallace et al., 2024), recent works have improved performance through negative samples and listwise ranking strategies (Bai et al., 2025). However, applying DPO to image inpainting is challenging due to data costs and limited exploration. InpaintDPO (Li et al., 2025b) attempted this task, but constructing high-quality mask-image pairs is expensive. Moreover, offline methods cannot explore beyond fixed datasets, limiting their effectiveness for open-ended editing. Therefore, online methods independent of paired data are a more promising direction.

**Online Preference Alignment.** Online alignment guides generation via reward models. DDPO (Black et al., 2024) treats denoising as a sequential decision problem using Proximal Policy Optimization (PPO). PrefPaint (Liu et al., 2024) applied this to inpainting for better aesthetic alignment. However, PPO requires a separate Critic network. This increases memory usage and training complexity. GRPO provides an efficient alternative by estimating baselines from group scores. It eliminates the need for a Critic model. Recently, DanceGRPO (Xue et al., 2025) and Flow-GRPO (Liu et al., 2025) extended GRPO to flow matching. They enable online exploration by converting ODE sampling to SDEs. MixGRPO (Li et al., 2025a) further improved this pipeline with hybrid sampling. Pref-GRPO (Wang et al., 2025) mitigated reward hacking in GRPO via pairwise pref-

erence rewards, enhancing training stability. Despite these advances in general generation, the application of GRPO and online DPO to image inpainting remains unexplored.

## 3. Preliminaries

### 3.1. Diffusion Models and Flow Matching

Diffusion models learn to reverse a forward noising process. Given clean data $z_0$, the forward process produces noisy samples via $q(z_t|z_0) = \mathcal{N}(z_t; \sqrt{\bar{\alpha}_t}z_0, (1-\bar{\alpha}_t)I)$, where $\bar{\alpha}_t$ controls the noise schedule and $I$ is the identity matrix. A neural network $\epsilon_\theta$ is trained to predict the added noise for iterative denoising.

Flow Matching learns a velocity field $v_\theta(z_t, t)$ to transport noise to data. The forward interpolation is $z_t = (1-t)z_0 + t\epsilon$, where $z_0$ is the data, $\epsilon \sim \mathcal{N}(0, I)$ is Gaussian noise, and $t \in [0, 1]$ is the timestep. Generation proceeds by solving $dz_t/dt = v_\theta(z_t, t)$ from $t = 1$ to $t = 0$.

To enable RL training, we inject exploration noise to convert the deterministic ODE into a stochastic process:

$$z_{t-\Delta t} = z_t - v_\theta(z_t, t) \cdot \Delta t + \eta\sqrt{\Delta t} \cdot \xi, \quad \xi \sim \mathcal{N}(0, I), \quad (1)$$

where $z_t$ is the current latent, $v_\theta(z_t, t)$ is the predicted velocity, $\Delta t > 0$ is the step size, $\eta$ controls the exploration noise scale, and $\xi$ is standard Gaussian noise. Since we integrate from $t = 1$ to $t = 0$, the velocity is subtracted. This yields a Gaussian transition with mean $\mu_\theta = z_t - v_\theta(z_t, t) \cdot \Delta t$ and variance $\eta^2 \Delta t$, giving tractable log-probability:

$$\log \pi_\theta(z_{t-\Delta t}|z_t) = -\frac{\|z_{t-\Delta t} - \mu_\theta\|^2}{2\eta^2\Delta t} + C, \quad (2)$$

where $\pi_\theta$ denotes the policy distribution and $C$ is a normalization constant independent of $\theta$.

### 3.2. Image Inpainting Formulation

Given an image $x \in \mathbb{R}^{H \times W \times 3}$ with height $H$ and width $W$, a binary mask $m \in \{0, 1\}^{H \times W}$ where $m_{i,j} = 1$ indicates masked pixels and $m_{i,j} = 0$ indicates preserved pixels, and a text prompt $c$, the goal is to generate $\hat{x}$ satisfying the consistency constraint: $\hat{x} \odot (1 - m) = x \odot (1 - m)$, where $\odot$ denotes element-wise multiplication. This preserves the unmasked regions unchanged while ensuring the filled area aligns with prompt $c$ and blends seamlessly with the context.

### 3.3. Reinforcement Learning Formulation

We cast inpainting as a Markov Decision Process $(\mathcal{S}, \mathcal{A}, \pi, r)$. The state $s_t = (z_t, m, c)$ comprises the noisy latent $z_t$, the mask $m$, and the prompt $c$. The action $a_t$ is the denoising step that predicts $z_{t-1}$ from $z_t$. The policy $\pi_\theta(z_{t-1}|z_t, m, c)$ is the diffusion or flow model parame-

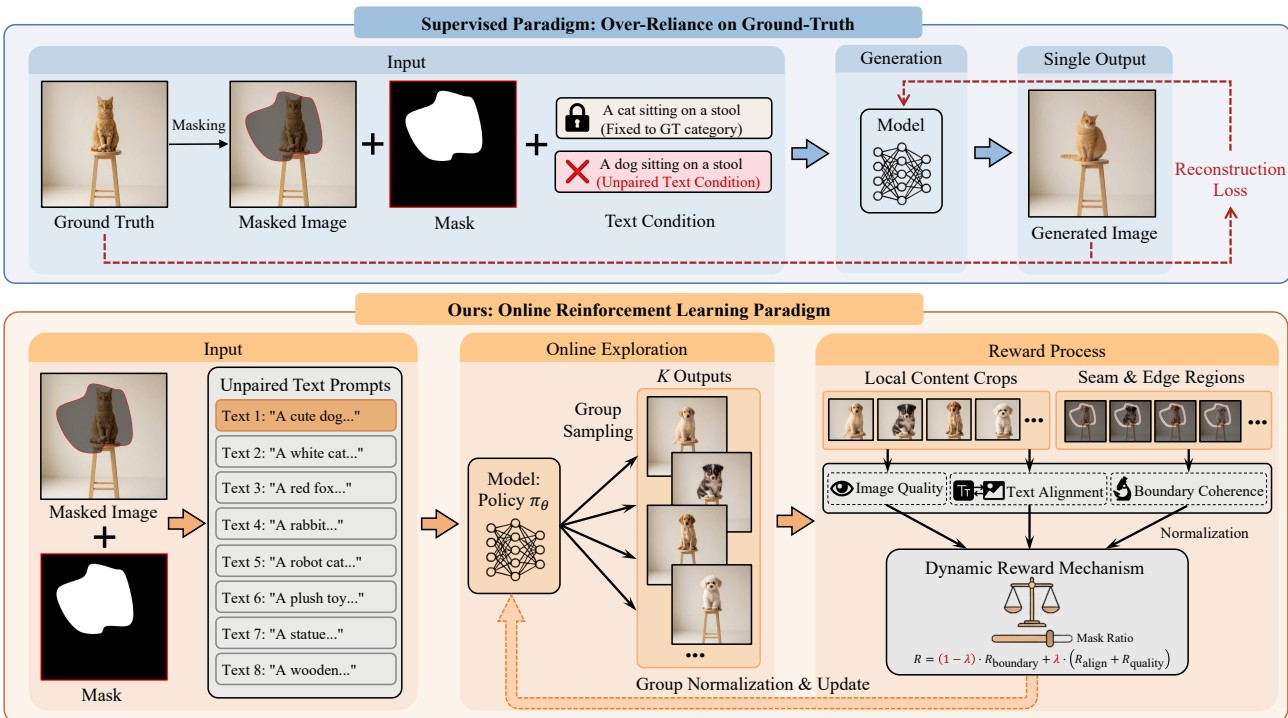

*Figure 2.* **Comparison of Training Paradigms. Top:** The traditional supervised paradigm suffers from an over-reliance on ground-truth reconstruction due to rigid text-image pairing. **Bottom:** Our proposed Online RL framework decouples this constraint by utilizing unpaired and online exploration. Guided by a dynamic reward mechanism, our method achieves diverse and aesthetically aligned generation beyond the limits of static datasets.

terized by $\theta$. The reward $r(z_0, m, c)$ evaluates the final generated image for quality and alignment.

GRPO eliminates the critic network by using group-relative baselines. For each input, the policy generates $K$ samples $\{z_0^{(1)}, \ldots, z_0^{(K)}\}$ with rewards $\{r_1, \ldots, r_K\}$. The advantage for sample $i$ is:

$$A_i = \frac{r_i - \bar{r}}{\sigma_r + \delta}, \quad \bar{r} = \frac{1}{K}\sum_{k=1}^{K} r_k, \quad \sigma_r = \sqrt{\frac{1}{K}\sum_{k=1}^{K}(r_k - \bar{r})^2}, \quad (3)$$

where $r_i$ is the reward for sample $i$, $\bar{r}$ is the group mean reward, $\sigma_r$ is the group standard deviation, and $\delta$ is a small constant for numerical stability. This enables learning from relative preferences without an explicit value function.

## 4. Methodology

As illustrated in Figure 2, we propose a novel Online RL framework for text-guided image inpainting. Unlike the traditional supervised paradigm (top), which constrains the model to reconstruct a single ground-truth image using fixed paired text and reconstruction loss, our approach (bottom) decouples the text-image correspondence to encourage diverse and creative generation. Specifically, the workflow

iterates through online exploration using unpaired prompts, reward evaluation via our dynamic reward mechanism, and policy update driven by GRPO or DPO. This iterative cycle enables the model to learn aesthetic alignment and seamless blending beyond the limitations of static datasets.

### 4.1. Online RL for Inpainting

**GRPO.** During sampling, we store the trajectory $\{z_T, \ldots, z_0\}$ produced by the generator and the corresponding log-probabilities $\{\log \pi_{\theta_{old}}(z_{t-1}|z_t)\}$ under the old policy. Training optimizes a clipped surrogate objective that constrains policy updates within a trust region, effectively stabilizing learning by preventing excessive divergence from the old policy:

$$\mathcal{L}_{GRPO} = -\mathbb{E}_t\left[\min\left(\rho_t A, \text{clip}(\rho_t, 1-\epsilon_c, 1+\epsilon_c)A\right)\right], \quad (4)$$

where $A$ is the group-relative advantage (Eq. 3), $\rho_t = \pi_\theta/\pi_{\theta_{old}}$ is the importance ratio, and $\epsilon_c$ is the clip range. The log-probability derives from the Gaussian transition:

$$\log \pi_\theta(z_{t-\Delta t}|z_t) = -\frac{\|z_{t-\Delta t} - \hat{z}_{t-\Delta t}\|^2}{2\eta^2\Delta t} + C, \quad (5)$$

where $\hat{z}_{t-\Delta t} = z_t - v_\theta(z_t, t) \cdot \Delta t$ is the deterministic prediction, $\eta$ is the exploration noise scale, $\Delta t > 0$ is the step size, and $C$ is a normalization constant.

**DPO.** Our online sampling naturally yields preference pairs by selecting the highest and lowest reward samples from each group. For a winner-loser pair $(y_w, y_l)$, the DPO loss is:

$$\mathcal{L}_{\text{DPO}} = -\log \sigma \left( \frac{\beta}{2} \left[ (e_\theta^l - e_{\text{ref}}^l) - (e_\theta^w - e_{\text{ref}}^w) \right] \right), \quad (6)$$

where $e_\theta^w$ and $e_\theta^l$ are prediction errors for winner and loser samples, $e_{\text{ref}}^w$ and $e_{\text{ref}}^l$ are reference model errors, $\sigma$ is the sigmoid function, and $\beta$ is the temperature. This encourages the model to decrease the winner's error relative to the loser. This provides a complementary objective when $K = 2$.

### 4.2. Boundary Gradient Consistency Score

To specifically quantify and eliminate the "stitching traces" mentioned in the introduction, we introduce the Boundary Gradient Consistency Score (BGCS). Standard aesthetic metrics often fail to detect high-frequency artifacts at the inpainting boundary. We address this by extracting inner and outer boundary regions via morphological operations:

$$M_{\text{in}} = m - \text{Erode}(m), \quad M_{\text{out}} = \text{Dilate}(m) - m, \quad (7)$$

where $m$ is the binary mask, $M_{\text{in}}$ denotes the newly generated pixels adjacent to the boundary, and $M_{\text{out}}$ denotes the preserved ground-truth pixels adjacent to the boundary. We then compute the mean gradient magnitude for each region:

$$\bar{G}_{\text{region}} = \frac{\sum_{i,j} \|\nabla I_{i,j}\| \cdot M_{\text{region}}^{i,j}}{\sum_{i,j} M_{\text{region}}^{i,j}}, \quad (8)$$

where $\nabla I_{i,j} = (G_x, G_y)$ is the Sobel gradient at pixel $(i, j)$. The boundary reward is defined as the negative absolute difference between these regions:

$$R_{\text{boundary}} = -|\bar{G}_{\text{in}} - \bar{G}_{\text{out}}|. \quad (9)$$

This formulation penalizes gradient discontinuity—both unnatural blur (low gradients) and noise (high gradients) are penalized when they deviate from the local texture characteristics of the unmasked background. We provide further verification of this metric's robustness against noise and blending operations in the Appendix (see Fig. 8).

### 4.3. Dynamic Reward Mechanism

Global evaluation metrics often encounter a "background dominance" issue in inpainting tasks because edits are typically local. As illustrated in Fig. 3, the normalized reward variance across methods increases with the mask coverage ratio. In quantitative analyses from previous studies (Ju et al., 2024b; Zhuang et al., 2024), the difference between top-performing methods and the lowest baselines is as small as 1% on certain metrics. This suggests that the unmasked

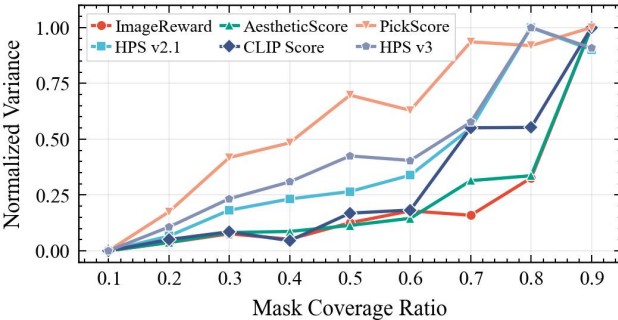

*Figure 3.* **Reward Sensitivity Analysis.** The normalized variance of reward scores exhibits a positive correlation with the mask coverage ratio, indicating that global metrics lack sufficient discriminative power for small-scale inpainting.

background dominates the global score. To address this issue, we formulate a dynamic reward mechanism as follows:

$$R = (1 - \lambda) \cdot R_{\text{boundary}} + \lambda \cdot (R_{\text{align}} + R_{\text{quality}}), \quad (10)$$

Here, $R_{\text{boundary}}$, $R_{\text{align}}$, and $R_{\text{quality}}$ denote the boundary, text alignment, and image quality scores, respectively. The term $\lambda = \frac{1}{HW} \sum_{i,j} m_{i,j}$ represents the mask coverage ratio, given the image resolution $H \times W$. This mechanism addresses the insensitivity of global evaluation metrics toward small editing regions. Since global scoring models often fail to distinguish between generation candidates when the mask is small, our mechanism automatically increases the proportion of $R_{\text{boundary}}$ when $\lambda$ is low, thereby prioritizing boundary consistency to ensure seamless integration. Conversely, as the mask area increases, the focus naturally shifts toward $R_{\text{align}}$ and $R_{\text{quality}}$ to encourage creative content generation. To ensure consistent score scales, each component is normalized within its generation group before the weighted combination. Additionally, we crop the input to the mask's bounding box to enhance metric discriminability.

## 5. Experiments

### 5.1. Experimental Settings

**Datasets and Benchmarks.** Following the research approach of BrushNet (Ju et al., 2024b), we sampled 24,000 stylistically diverse images from the LAION dataset (Schuhmann et al., 2022). We adopted scripts to randomly generate masks, and leveraged multimodal large language models to arbitrarily generate unrestricted textual prompts according to the original images and masks, thereby completing the construction of the training set. We prepared two resolution groups for training: $1024 \times 1024$ for Flux.1-fill and $512 \times 512$ for SD1.5-inpainting. Our evaluation employs three benchmarks to assess generalization across scales. We first utilize **FluxBench** (Black Forest Labs, 2024), comprising 50 high-resolution images derived from the official Black Forest Labs suite. Furthermore, to ensure alignment

*Table 1.* **Quantitative comparison across three benchmarks.** Methods are grouped into the SD1.5 series (first six rows) and the Flux.1 series (last three rows). Note that baselines BrushNetX and PowerPaint v2 are based on the BrushNet architecture, while PrefPaint uses the PPO algorithm. **Red**, **Blue**, and **bold** denote the **1st**, **2nd**, and **3rd** best results.

| Metrics
Models | Image Quality | | | | | Mask Preservation | | | Text Align | Boundary |
|---|---|---|---|---|---|---|---|---|---|---|
| | $IR_{\times 10^2}\uparrow$ | $HPSv2.1_{\times 10^2}\uparrow$ | AS$\uparrow$ | HPSv3$\uparrow$ | PickScore$\uparrow$ | PSNR$\uparrow$ | $LPIPS_{\times 10^2}\downarrow$ | $MSE_{\times 10^2}\downarrow$ | $CLIP_{\times 10^2}\uparrow$ | BGCS$\uparrow$ |
| **FluxBench** | | | | | | | | | | |
| BrushNetX | -21.38 | 26.58 | 5.641 | 3.885 | 20.91 | 5.298 | 8.50 | 32.33 | 24.48 | -0.0298 |
| PowerPaint v2 | -24.99 | 26.66 | 5.563 | 3.808 | 21.01 | 5.326 | 8.74 | 35.17 | 25.23 | -0.0320 |
| PrefPaint | -127.7 | 24.57 | 5.042 | -1.651 | 19.69 | 6.398 | 7.44 | 28.17 | 21.74 | -0.0326 |
| SD1.5 | -83.33 | 25.75 | 5.311 | 0.823 | 20.16 | 5.576 | 8.21 | 31.98 | 23.52 | -0.0404 |
| SD1.5 (SFT) | -73.32 | 26.00 | 5.375 | 0.550 | 20.29 | 4.426 | 9.64 | 42.03 | 24.03 | -0.2634 |
| **SD1.5 (Ours)** | -26.17 | 26.48 | 5.546 | 2.695 | 20.85 | 5.325 | 8.57 | 33.85 | 24.62 | -0.0295 |
| Flux.1 | -32.53 | 26.06 | 5.326 | 3.099 | 20.94 | 6.993 | 6.55 | 24.35 | 24.78 | -0.0369 |
| Flux.1 (SFT) | -34.57 | 26.35 | 5.343 | 2.753 | 20.98 | 6.535 | 7.67 | 29.42 | 24.92 | -0.0768 |
| **Flux.1 (Ours)** | 31.41 | 27.14 | 5.684 | 4.461 | 21.44 | 6.278 | 7.17 | 27.77 | 25.61 | -0.0163 |
| **EditBench** | | | | | | | | | | |
| BrushNetX | -30.05 | 27.10 | 5.421 | 1.987 | 20.92 | 5.480 | 17.60 | 31.06 | 27.97 | -0.0356 |
| PowerPaint v2 | -45.56 | 26.98 | 5.393 | 2.038 | 20.94 | 5.607 | 17.20 | 30.47 | 27.69 | -0.0162 |
| PrefPaint | -107.7 | 25.25 | 4.657 | -3.921 | 19.76 | 5.994 | 16.17 | 28.68 | 25.32 | -0.0212 |
| SD1.5 | -82.84 | 25.96 | 4.854 | -2.571 | 20.11 | 5.480 | 16.54 | 31.20 | 26.53 | -0.0240 |
| SD1.5 (SFT) | -98.37 | 25.69 | 4.870 | -3.276 | 19.99 | 5.335 | 17.55 | 34.01 | 26.70 | -0.1119 |
| **SD1.5 (Ours)** | -40.63 | 26.37 | 5.164 | 0.317 | 20.83 | 5.481 | 17.44 | 31.09 | 27.50 | -0.0156 |
| Flux.1 | -33.36 | 26.01 | 4.744 | -2.040 | 20.49 | 7.446 | 14.47 | 20.91 | 27.33 | -0.0085 |
| Flux.1 (SFT) | -36.23 | 25.85 | 4.780 | -1.932 | 20.44 | 7.307 | 15.16 | 22.58 | 27.47 | -0.0326 |
| **Flux.1 (Ours)** | 8.98 | 27.18 | 5.503 | 2.779 | 21.11 | 6.970 | 14.99 | 22.05 | 28.14 | -0.0042 |
| **BrushBench** | | | | | | | | | | |
| BrushNetX | 55.42 | 26.55 | 5.824 | 1.099 | 20.95 | 5.440 | 8.96 | 33.85 | 25.72 | -0.2222 |
| PowerPaint v2 | 59.42 | 26.56 | 5.915 | 1.446 | 21.16 | 5.998 | 8.53 | 29.89 | 26.26 | -0.1364 |
| PrefPaint | 30.75 | 25.86 | 5.494 | -2.124 | 20.33 | 5.773 | 8.63 | 30.80 | 25.39 | -0.1539 |
| SD1.5 | 38.40 | 26.05 | 5.558 | -1.296 | 20.46 | 5.565 | 8.80 | 32.27 | 25.63 | -0.1643 |
| SD1.5 (SFT) | 38.81 | 26.08 | 5.546 | -1.165 | 20.51 | 5.567 | 8.51 | 37.96 | 25.54 | -0.1864 |
| **SD1.5 (Ours)** | 53.86 | 26.49 | 5.810 | 0.962 | 20.97 | 5.456 | 8.86 | 32.52 | 25.90 | -0.1157 |
| Flux.1 | 48.64 | 26.15 | 5.629 | 0.353 | 20.94 | 7.937 | 7.14 | 19.99 | 25.25 | -0.1250 |
| Flux.1 (SFT) | 49.17 | 26.16 | 5.617 | 0.068 | 20.95 | 7.944 | 7.08 | 19.87 | 25.27 | -0.1468 |
| **Flux.1 (Ours)** | 55.03 | 26.53 | 5.798 | 0.871 | 20.98 | 7.853 | 7.13 | 20.22 | 25.76 | -0.1114 |

with the BrushNet baseline, we adopt **EditBench** (Wang et al., 2023) (240 images at $1024 \times 1024$) and **Brush-Bench** (Ju et al., 2024b) (600 images at $512 \times 512$).

**Baselines.** We select **PowerPaint v2** (Zhuang et al., 2024) and **BrushNetX** (Li et al., 2024), which have gained widespread popularity among developers in the open-source community due to their exceptional generation quality and robustness. We also select **PrefPaint** (Liu et al., 2024), which utilizes the PPO algorithm for image inpainting. Throughout the experiment, we did not perform any additional fine-tuning, hyperparameter tuning, or secondary training on these baseline models. These methods, while achieving high-quality results, incur significant additional costs due to architectural extensions and reliance on large-scale external training data. Additionally, we include the following: (1) the pre-trained **SD1.5-inpainting** (hereafter SD1.5) and the 12B-parameter **Flux.1-fill** (hereafter Flux.1); and (2) their Supervised Fine-Tuning (SFT) counterparts, which were fine-tuned using the identical dataset to our method.

**Evaluation Metrics.** Following established conventions and complementing metrics from prior prominent works, we conducted a comprehensive evaluation using 10 distinct metrics across four dimensions. For image quality, we em-

ployed ImageReward (**IR**) (Xu et al., 2023), **HPSv2.1** (Wu et al., 2023), Aesthetic Score (**AS**) (Schuhmann et al., 2022), **HPSv3** (Ma et al., 2025), and **PickScore** (Kirstain et al., 2023). Regarding mask preservation, we utilized Peak Signal-to-Noise Ratio (**PSNR**), Learned Perceptual Image Patch Similarity (**LPIPS**) (Zhang et al., 2018), and Mean Squared Error (**MSE**) to quantify consistency in unmasked regions. Text alignment was measured via CLIP Similarity (**CLIP**) (Radford et al., 2021). Finally, to evaluate the naturalness of transition regions, we assessed the Boundary Gradient Consistency Score (**BGCS**).

**Implementation Details.** All experiments were conducted using DanceGRPO (Xue et al., 2025) within the FastVideo (Zhang et al., 2025) framework. DanceGRPO uniquely employs a shared architecture for both the Actor and Reference models to maximize VRAM efficiency, while leveraging stricter clipping constraints to prevent policy over-divergence. Consequently, the computational memory overhead is comparable to SFT, incurring only the additional cost of reward model evaluation. We conducted training on 4× NVIDIA H20 (96GB) GPUs. For the Flux model with 12 billion parameters, when using bfloat16 precision and offloading computations to the CPU, the peak memory footprint during training can be reduced to approximately 180 GB, with the full training cycle taking about four days.

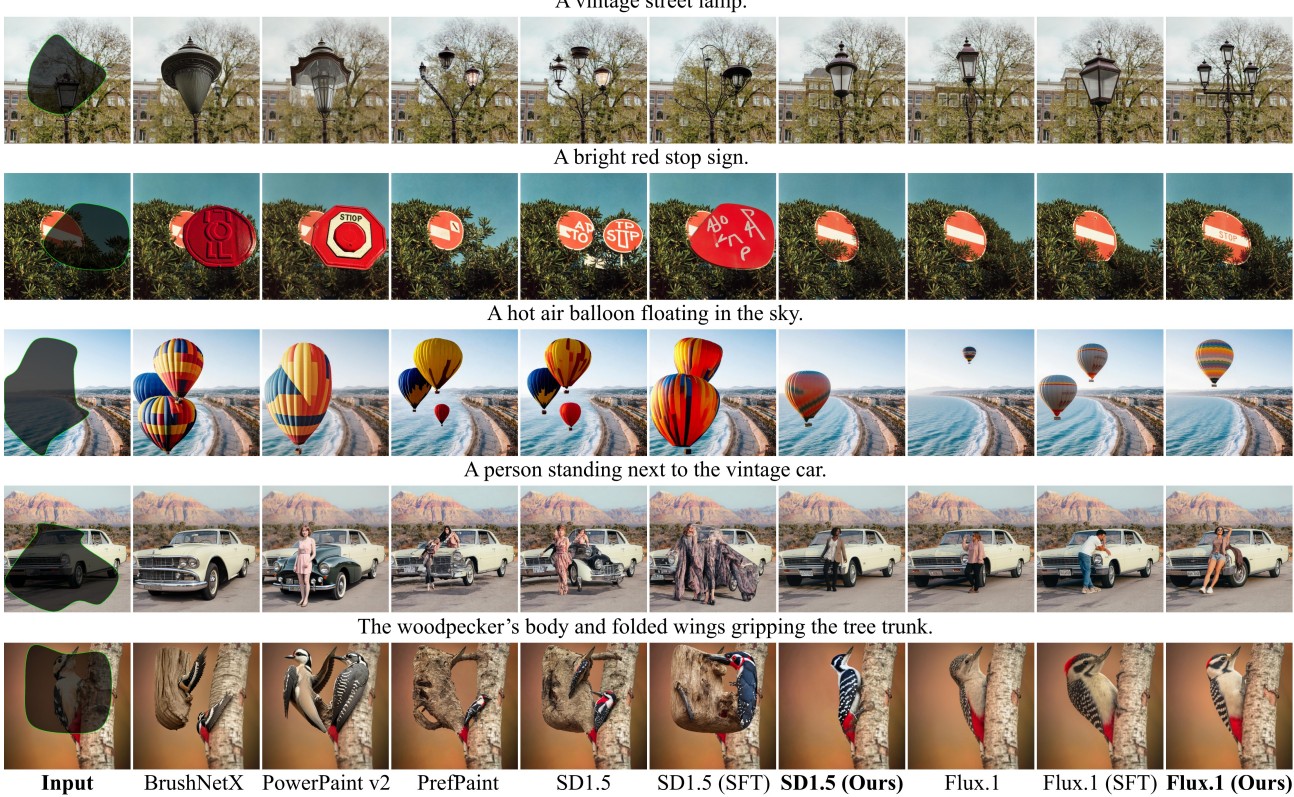

*Figure 4.* **Qualitative comparison with state-of-the-art methods.** The leftmost column displays the masked input images, and the corresponding text prompts are provided above each row. Our method demonstrates superior visual appeal.

## 5.2. Quantitative Analysis

Table 1 presents our quantitative analysis. Given the differing pre-training resolutions, we adopt a grouped evaluation strategy. We focus on the high-resolution FluxBench and EditBench for the Flux.1 series (last three rows), while examining BrushBench ($512 \times 512$) for the SD1.5 series (first six rows).

Our method demonstrates strong cross-scale generalization. Flux.1 (Ours), trained at $1024 \times 1024$, achieves state-of-the-art results on FluxBench and EditBench. Notably, it also outperforms the base model on the lower-resolution BrushBench. Similarly, SD1.5 (Ours) exhibits substantial performance improvements on high-resolution benchmarks, despite being trained at only $512 \times 512$.

Baselines such as BrushNetX and PowerPaint v2 have been widely adopted by developers in the open-source community. These methods utilize external BrushNet architectures, introducing adapters that account for approximately 70% to over 100% of the base UNet parameters and relying on massive, TB-scale external training datasets. In contrast, our method applies direct fine-tuning to the base model, adding no extra parameters during inference. Consequently, on BrushBench ($512 \times 512$), our approach significantly enhances the capa-

bilities of the SD1.5 model, achieving performance second only to these resource-intensive state-of-the-art baselines.

Regarding training paradigms, the current dataset size appears insufficient for SFT, resulting in model instability and performance metrics comparable to the baseline. In contrast, our method delivers substantial improvements in image quality. Specifically, since our approach eliminates the strict constraint of ground-truth reconstruction to foster solution diversity, a corresponding decline in mask preservation is observed.

## 5.3. Qualitative Analysis

To intuitively evaluate robustness in complex scenarios, Figure 4 presents results under random mask settings. All images used for qualitative analysis in this paper are real-world photographs sourced from Unsplash[2], ensuring the validity and generalizability of our visual comparisons. This task becomes challenging when masks partially cover objects and leave limited semantic cues. Following the input column, the next six columns belong to the SD1.5 series. While other models fail to capture the context, our method

---

[2]Unsplash (https://unsplash.com/) offers authentic real-world images, free for all uses without attribution.

accurately completes the semantics. The last three columns display the Flux series. Although the base and SFT models generate reasonable structures, our method further improves visual appeal. It produces images that better align with aesthetic preferences.

### 5.4. Ablation Studies

#### 5.4.1. EFFECT OF REWARD COMPONENTS

To validate the necessity of each component in our dynamic reward mechanism, we conducted an ablation study. The baselines are configured as follows: "*Base*" denotes the original model without RL training; "*w/o Boundary*" utilizes only HPSv2.1 and CLIP with equal weights ($1:1$), a common setting in T2I RL; "*w/o Dynamic*" introduces the BGCS but maintains fixed $1:1:1$ weights, independent of the mask ratio; and "*Ours*" employs the full dynamic reward mechanism to adaptively balance the boundary and global rewards based on the mask ratio.

*Table 2.* **Ablation study on reward components. Red** and **Blue** denote the **1st** and **2nd** best results.

| Metrics | Image Quality | | | | | Boundary |
|---|---|---|---|---|---|---|
| Models | $\text{IR}_{\times 10^2}\uparrow$ | $\text{HPSv2.1}_{\times 10^2}\uparrow$ | AS↑ | HPSv3↑ | PickScore↑ | BGCS↑ |
| **FluxBench** Base | -32.53 | 26.06 | 5.326 | 3.099 | 20.94 | -0.0369 |
| w/o Boundary | **38.73** | 26.98 | **5.646** | **4.238** | 21.32 | -0.0304 |
| w/o Dynamic | 29.14 | **27.09** | 5.642 | 4.193 | **21.38** | **-0.0183** |
| **Ours** | **31.41** | **27.14** | **5.684** | **4.461** | **21.44** | **-0.0163** |
| **EditBench** Base | -33.36 | 26.01 | 4.744 | -2.040 | 20.49 | -0.0085 |
| w/o Boundary | -4.51 | 26.97 | 5.408 | 2.1181 | 20.96 | -0.0087 |
| w/o Dynamic | **4.66** | **27.00** | **5.509** | **2.2390** | **21.01** | **-0.0051** |
| **Ours** | **8.98** | **27.18** | 5.503 | **2.7792** | **21.11** | **-0.0042** |
| **BrushBench** Base | 48.64 | 26.15 | 5.629 | 0.3534 | 20.94 | -0.1250 |
| w/o Boundary | 53.29 | **26.51** | 5.761 | **0.8346** | **20.95** | -0.1232 |
| w/o Dynamic | **54.71** | 26.49 | **5.786** | 0.8150 | 20.93 | **-0.1169** |
| **Ours** | **55.03** | **26.53** | **5.798** | **0.8712** | **20.98** | **-0.1114** |

As shown in Table 2, quantitative analysis indicates that employing RL alone significantly enhances image quality scores but fails to improve the BGCS metric, occasionally even leading to slight degradation. The incorporation of the BGCS term, even with static weights, yields substantial improvements in boundary quality, demonstrating its effectiveness in penalizing structural inconsistencies. Crucially, our adaptive weighting mechanism achieves the best performance in both image quality and boundary fidelity, thereby demonstrating the effectiveness of both the BGCS and the dynamic reward mechanism.

Qualitatively, Figure 5 illustrates the visual impact of different reward components on generative details. While RL generally enhances visual quality, configurations lacking boundary constraints ("*Base*" and "*w/o Boundary*") tend to produce overly smoothed textures. This deficiency is particularly evident in high-frequency texture regions such as "grass", where fine structural details are blurred or homogenized, leading to perceptible artifacts and a loss of textural

authenticity. The introduction of our proposed BGCS effectively restores the fine granularity of grass blades and other intricate patterns by penalizing gradient discontinuities and enforcing local consistency. Finally, our full method ("*Ours*") generates results that best align with aesthetic expectations while effectively eliminating visual seams.

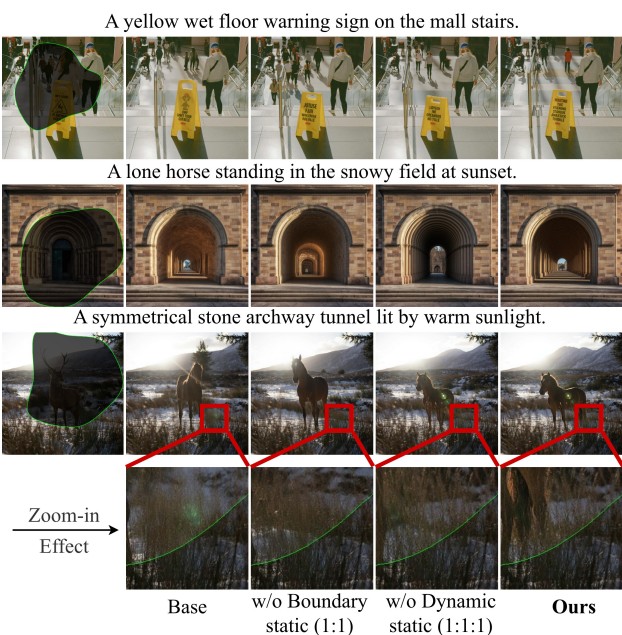

*Figure 5.* **Visual ablation on reward components.** Zoomed-in regions (red boxes) highlight the grassy terrain, demonstrating progressive improvement in boundary coherence and fine-grained texture generation.

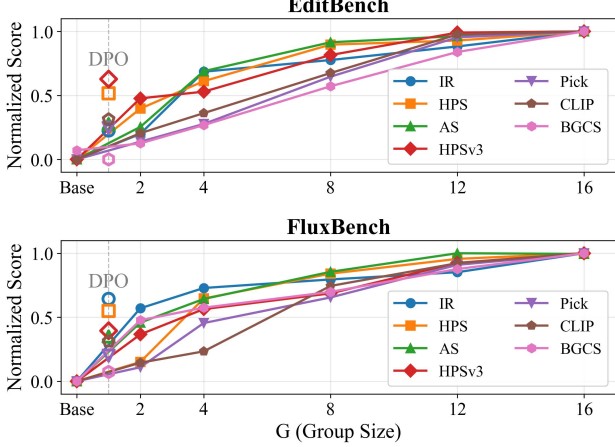

*Figure 6.* **Impact of group size $K$ and comparison with DPO.** Larger group sizes yield consistent improvements for all metrics.

#### 5.4.2. EFFECT OF GROUP SIZE $G$

We further investigate the impact of group size $G$ during GRPO training, where $G$ is selected from $\{2, 4, 8, 12, 16\}$.

As illustrated by the normalized curves in Figure 6, all evaluation metrics exhibit a monotonic upward trend as $G$ increases. This indicates that larger sampling groups yield superior performance. However, the sampling time increases linearly with $G$. To balance performance and training efficiency, we adopt $G = 12$ as our default setting. We also observe that Online DPO serves as an efficient alternative in resource-constrained scenarios. By sampling only two images to form a preference pair, Online DPO achieves performance comparable to GRPO with $G = 4$ while reducing the sampling time by half. Detailed comparisons are provided in Table 5 in the Appendix.

### 5.5. Human Evaluation

To comprehensively evaluate the subjective quality of our proposed inpainting method, we conducted a large-scale human evaluation study. We recruited a total of 76 participants and performed a double-blind pairwise preference comparison experiment across 10 diverse real-world image inpainting scenarios. Throughout the entire testing process, the identity of all models was fully concealed from participants to eliminate any potential bias and ensure the objectivity of the evaluation results. The aggregated preference votes of each model across all real-world scenarios are summarized in Table 3.

*Table 3.* Aggregated human preference votes across real-world inpainting scenarios

| Model | Flux.1 (Ours) | BrushNetX | PowerPaint v2 | PrefPaint | SD1.5 (Ours) |
|---|---|---|---|---|---|
| Total Votes | **410** | 122 | 98 | 27 | 103 |

Experimental results demonstrate that our proposed method based on the Flux.1 backbone achieves the highest total preference votes, significantly outperforming all competing methods. Meanwhile, the SD1.5 backbone model optimized by our method obtains preference votes comparable to those of state-of-the-art methods including BrushNetX and PowerPaint v2, further validating the effectiveness of our proposed optimization strategy.

## 6. Conclusion

We present the first online Reinforcement Learning framework to explore GRPO and DPO for image inpainting. Our method decouples the strict text-image pairing constraint of traditional supervised paradigms. To address "stitching traces" and the "background dominance" issue, we introduce a dynamic reward mechanism incorporating the BGCS. Experiments demonstrate that our approach significantly enhances the visual quality of Stable Diffusion and FLUX models without introducing additional inference parameters.

**Limitations.** Our dynamic reward mechanism improves

performance on mainstream benchmarks with minimal overhead. Yet, quantifying human perception of inpainting boundaries and aesthetics remains challenging. Future work should explore boundary metrics with higher perceptual discriminability and finer-grained reward designs to better align objective evaluation with human aesthetic intuition.

## Impact Statement

This work aims to advance the field of machine learning by improving the controllability and quality of image inpainting. High-performance inpainting empowers creative workflows, benefiting artists and designers by automating complex editing tasks. However, this progress introduces risks of visual forgery. Advanced inpainting tools could be misused to tamper with documents, remove individuals from evidence, violate privacy, or generate disinformation. To mitigate these risks, the community must develop robust forensic detection tools capable of identifying infilled regions. We also advocate for the adoption of digital watermarking standards. These standards effectively identify and track AI-modified content, ensuring responsible and transparent usage.

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

# A. Appendix

## A.1. Experimental Settings

To ensure reproducibility and fairness, we unify the sampling and training settings. The SD1.5-inpainting series employs 50 denoising steps for both training and inference. In contrast, the Flux.1-fill series utilizes 20 steps. For the GRPO configuration, the default group size is set to $G = 12$, generating 12 independent samples per prompt.

## A.2. Benchmark Visualizations

To provide an intuitive visualization of the evaluation benchmarks, this section presents representative samples from the three datasets, as shown in Figure 7. **FluxBench** (Black Forest Labs, 2024) features single-block masks and consists of 50 high-resolution images with varying dimensions. **EditBench** (Wang et al., 2023) utilizes single-block masks and comprises 240 images at $1024 \times 1024$ resolution, including an equal split of 120 real and 120 synthetic images. **BrushBench** (Ju et al., 2024b) employs object-shaped masks and contains 600 images ($512 \times 512$) sourced from the LAION dataset (Schuhmann et al., 2022).

## A.3. Specifications of Evaluation Metrics

Table 4 provides a detailed overview of the evaluation metrics employed in our experiments, including their input modalities, output formats, and backbone architectures where applicable.

*Table 4.* Specifications of Evaluation Metrics used in experiments.

| Model | Input | Output | Source | Note |
|---|---|---|---|---|
| **ImageReward** (Xu et al., 2023) | Image + Text | Scalar | HuggingFace | - |
| **HPS v2.1** (Wu et al., 2023) | Image + Text | Scalar | GitHub | - |
| **Aesthetic Score** (Schuhmann et al., 2022) | Image | Scalar | GitHub | ViT-L-14 |
| **HPS v3** (Ma et al., 2025) | Image + Text | Scalar | GitHub | Qwen2-VL-7B |
| **PickScore** (Kirstain et al., 2023) | Image + Text | Scalar | HuggingFace | - |
| **CLIP** (Radford et al., 2021) | Image + Text | Scalar | HuggingFace | ViT-L-14 |

## A.4. Robustness Analysis of Boundary Metric

A quantitative sensitivity analysis on 100 random inpainting tasks validates the reliability of BGCS. As shown in Fig. 8, the metric exhibits a monotonic increase with respect to both boundary Gaussian noise variance (left) and blending intensity (right). This confirms that BGCS effectively penalizes both high-frequency discontinuities and gradient mismatches caused by over-smoothing.

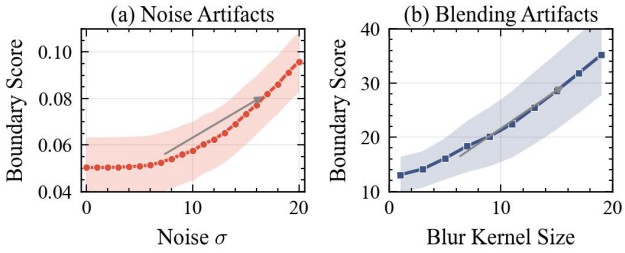

*Figure 8.* **Sensitivity of BGCS to boundary artifacts.** The score increases monotonically with noise variance (Left) and blending intensity (Right). Higher values indicate poorer consistency, demonstrating the metric's capability to detect both high-frequency noise and blurriness.

### A.4.1. ROBUSTNESS TO SPECIAL MASK TYPES

Our method employs script-generated free-form masks that cover diverse complex scenarios, including arbitrary proportions and irregular shapes. This design closely mimics the manual mask brushing behavior of real-world users. We identify a critical edge case: when the inpainted region exactly aligns with a natural semantic boundary, large gradient discontinuities are perceptually valid and should not be penalized by our boundary consistency loss. To address this, we propose a simple yet effective solution: dilating the input semantic segmentation masks outward by a small number of pixels. We further validate the robustness of our approach on two challenging mask distributions: semantic segmentation masks that precisely follow object contours, and highly complex, irregular masks with intricate hole patterns. Qualitative results in Figure 9(A) and (B) demonstrate that our method consistently achieves superior texture coherence and boundary naturalness across all mask types.

### A.4.2. ROBUSTNESS TO VAGUE AND OPEN-ENDED PROMPTS

A key advantage of text-guided image inpainting is its ability to handle ambiguous user instructions. We evaluate our framework's robustness under vague and open-ended prompts, where the desired content is not explicitly specified. As shown in Figure 9(C), while the base SFT model leverages its extensive pre-training knowledge to produce plausible results, our method further improves generation quality by incorporating human preference alignment. Our model consistently generates more semantically coherent and visually pleasing outputs even when given minimal textual guidance.

### A.4.3. DIVERSITY UNDER IDENTICAL PROMPTS

While our primary focus is on instruction-level diversity (supporting different text instructions for the same masked region), we also evaluate the intra-prompt diversity of our model. This corresponds to the real-world scenario where users may want multiple alternative results for the same editing request. As shown in Figure 9(D), our model exhibits significantly higher generation diversity across different random seeds compared to baseline methods, while maintaining high visual quality and semantic consistency with the input prompt.

## A.5. Comprehensive Quantitative Analysis

We present more detailed quantitative evaluation results in Table 5.

**FluxBench**

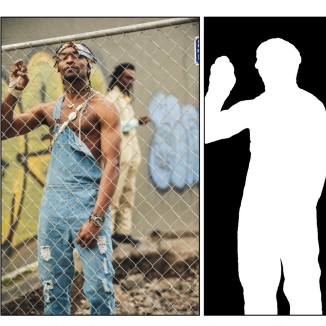 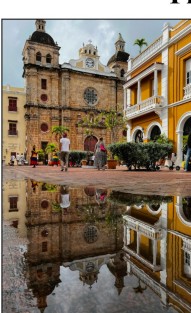 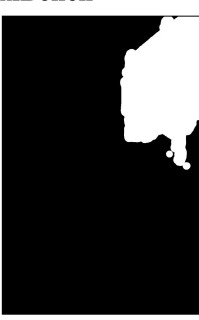 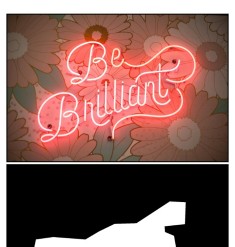 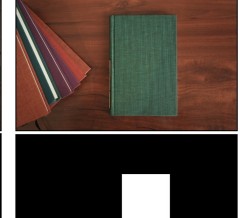

1. a black man wearing yellow, jeans overalls
2. a building
3. the words "Black Forest Labs", and in the next line "The diary"
4. a sign that says "different"

**EditBench**

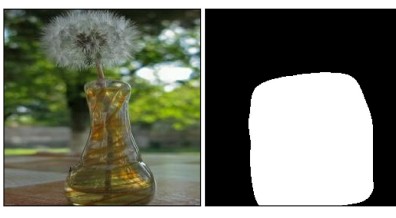 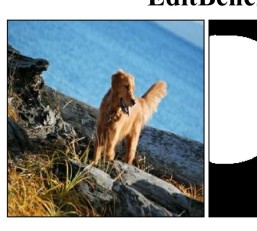 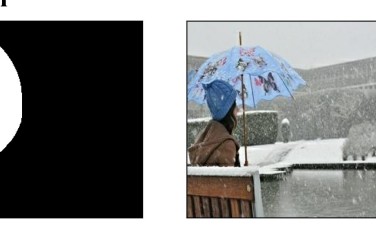 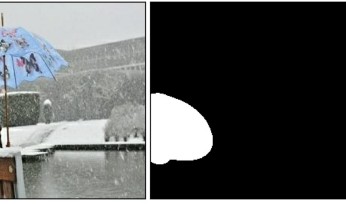

1. a dandelion in an amber vase with trees in the background
2. a brown dog standing to the left of a dark angular rock on the beach
3. a woman in a beige jacket sitting in a bench in the snow

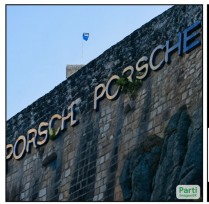 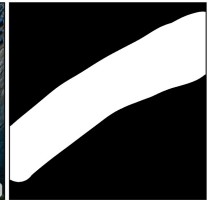 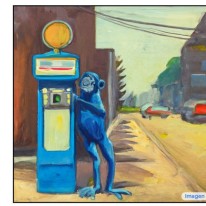 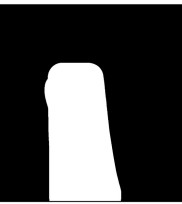 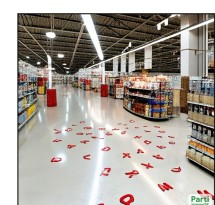 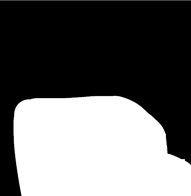

1. two copies of beige and navy colored letters "porsche" on the wall of a castle on top of a mountain
2. the oil painting of a blue monkey leaning on a coin machine next to a street
3. a few flat-shaped letters scattered on the floor in a spacious grocery store

**BrushBench**

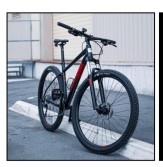 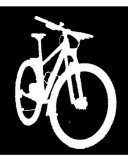 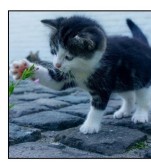 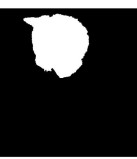 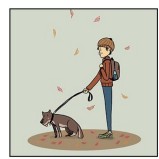 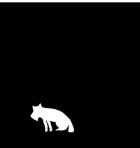 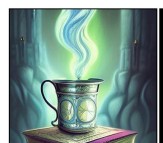 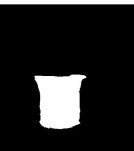

1. a black and red mountain bike parked on the side of a building
2. a kitten is playing with a flower
3. a boy walking his dog in the park
4. a painting of a cup on top of books

*Figure 7.* **Examples of Benchmarks.** The text prompts below correspond to the images from left to right. Each sample consists of a text prompt and an image pair, where the left image represents the input and the right image represents the corresponding binary mask. The model is tasked with inpainting the white regions of the mask based on the text prompt.

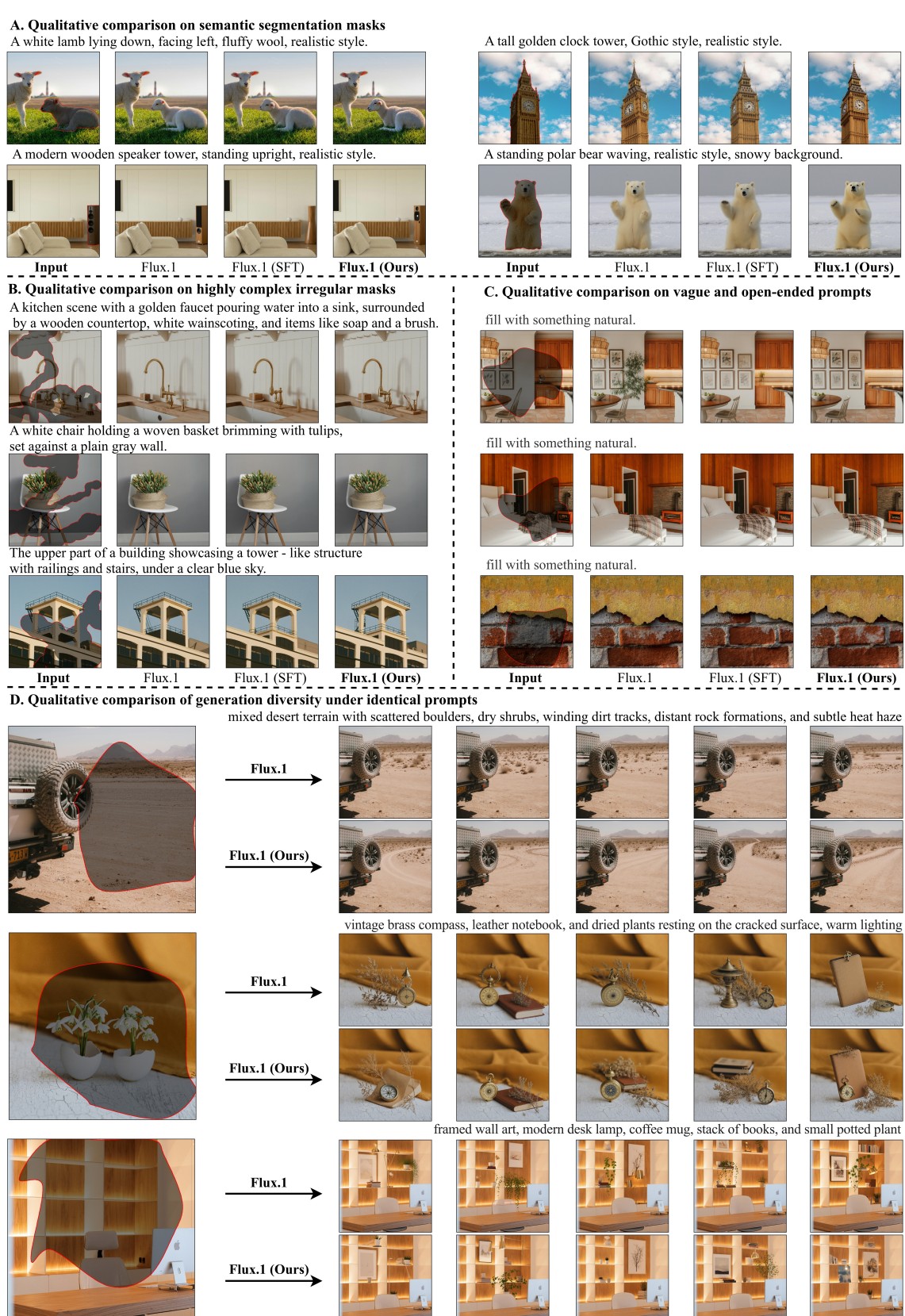

*Figure 9.* Comprehensive ablation studies on model robustness and diversity.

*Table 5.* **Comprehensive quantitative comparison and ablation analysis.** We report metrics for baselines, training paradigms (SFT, DPO), and variations of our method (denotes GRPO group size). Configurations: '*w/o Boundary*": HPSv2.1 and CLIP (1 : 1); '*w/o Dynamic*": adds BGCS with fixed weights (1 : 1 : 1); '*w/o Cropping*": rewards computed on full image without mask-based cropping; '*Ours*": full dynamic reward mechanism balancing boundary and global scores via mask ratio.

| | Metrics | Image Quality | | | | | Mask Preservation | | | Text Align | Boundary |
| | Models | IR↑ | HPSv2.1↑ | AS↑ | HPSv3↑ | PickScore↑ | PSNR↑ | LPIPS↓ | MSE↓ | CLIP↑ | BGCS↑ |
|---|---|---|---|---|---|---|---|---|---|---|---|
| **FluxBench** | BrushNetX | -0.2138 | 0.2658 | 5.641 | 3.885 | 20.91 | 5.298 | 0.0850 | 0.3233 | 0.2448 | -0.0298 |
| | PowerPaint v2 | -0.2499 | 0.2666 | 5.563 | 3.808 | 21.01 | 5.326 | 0.0874 | 0.3517 | 0.2523 | -0.0320 |
| | PrefPaint | -1.277 | 0.2457 | 5.042 | -1.651 | 19.69 | 6.398 | 0.0744 | 0.2817 | 0.2174 | -0.0326 |
| | SD1.5 | -0.8333 | 0.2575 | 5.311 | 0.823 | 20.16 | 5.576 | 0.0821 | 0.3198 | 0.2352 | -0.0404 |
| | SD1.5 (SFT) | -0.7332 | 0.2600 | 5.375 | 0.550 | 20.29 | 4.426 | 0.0964 | 0.4203 | 0.2403 | -0.2634 |
| | SD1.5 (DPO) | -0.6324 | 0.2619 | 5.407 | 1.827 | 20.43 | 5.488 | 0.0820 | 0.3270 | 0.2405 | -0.0409 |
| | SD1.5 (G=12 w/o Boundary) | -0.4054 | 0.2644 | 5.519 | 2.476 | 20.73 | 5.387 | 0.0845 | 0.3338 | 0.2465 | -0.0391 |
| | SD1.5 (G=12 Ours) | -0.2617 | 0.2648 | 5.546 | 2.695 | 20.85 | 5.325 | 0.0857 | 0.3385 | 0.2462 | -0.0295 |
| | Flux.1 | -0.3253 | 0.2606 | 5.326 | 3.099 | 20.94 | 6.993 | 0.0655 | 0.2435 | 0.2478 | -0.0369 |
| | Flux.1 (SFT) | -0.3457 | 0.2635 | 5.343 | 2.753 | 20.98 | 6.535 | 0.0767 | 0.2942 | 0.2492 | -0.0768 |
| | Flux.1 (DPO) | 0.1577 | 0.2668 | 5.451 | 3.680 | 21.05 | 6.833 | 0.0688 | 0.2510 | 0.2506 | -0.0352 |
| | Flux.1 (G=12 w/o Boundary) | 0.3873 | 0.2698 | 5.646 | 4.238 | 21.32 | 6.462 | 0.0700 | 0.2688 | 0.2559 | -0.0304 |
| | Flux.1 (G=12 w/o Dynamic) | 0.2914 | 0.2709 | 5.642 | 4.193 | 21.38 | 6.376 | 0.0732 | 0.2724 | 0.2559 | -0.0183 |
| | Flux.1 (G=12 w/o Cropping) | 0.3158 | 0.2714 | 5.675 | 4.331 | 21.43 | 6.295 | 0.0734 | 0.2758 | 0.2561 | -0.0166 |
| | Flux.1 (G=2 Ours) | 0.1024 | 0.2623 | 5.490 | 3.638 | 21.00 | 6.913 | 0.0669 | 0.2469 | 0.2491 | -0.0257 |
| | Flux.1 (G=4 Ours) | 0.2215 | 0.2679 | 5.556 | 3.929 | 21.19 | 6.635 | 0.0690 | 0.2617 | 0.2499 | -0.0234 |
| | Flux.1 (G=8 Ours) | 0.2716 | 0.2701 | 5.632 | 4.110 | 21.30 | 6.478 | 0.0703 | 0.2671 | 0.2545 | -0.0205 |
| | Flux.1 (G=12 Ours) | 0.3141 | 0.2714 | 5.684 | 4.461 | 21.44 | 6.278 | 0.0717 | 0.2777 | 0.2561 | -0.0163 |
| | Flux.1 (G=16 Ours) | 0.4253 | 0.2719 | 5.682 | 4.572 | 21.49 | 6.215 | 0.0742 | 0.2778 | 0.2568 | -0.0134 |
| **EditBench** | BrushNetX | -0.3005 | 0.2710 | 5.421 | 1.987 | 20.92 | 5.480 | 0.1760 | 0.3106 | 0.2797 | -0.0356 |
| | PowerPaint v2 | -0.4556 | 0.2698 | 5.393 | 2.038 | 20.94 | 5.607 | 0.1720 | 0.3047 | 0.2769 | -0.0162 |
| | PrefPaint | -1.077 | 0.2525 | 4.657 | -3.921 | 19.76 | 5.994 | 0.1617 | 0.2868 | 0.2532 | -0.0212 |
| | SD1.5 | -0.8284 | 0.2596 | 4.854 | -2.571 | 20.11 | 5.480 | 0.1654 | 0.3120 | 0.2653 | -0.0240 |
| | SD1.5 (SFT) | -0.9837 | 0.2569 | 4.870 | -3.276 | 19.99 | 5.335 | 0.1755 | 0.3401 | 0.2670 | -0.1119 |
| | SD1.5 (DPO) | -0.6542 | 0.2605 | 5.053 | -0.926 | 20.39 | 5.479 | 0.1699 | 0.3119 | 0.2701 | -0.0222 |
| | SD1.5 (G=12 w/o Boundary) | -0.4850 | 0.2638 | 5.122 | -0.052 | 20.63 | 5.477 | 0.1718 | 0.3112 | 0.2742 | -0.0217 |
| | SD1.5 (G=12 Ours) | -0.4063 | 0.2637 | 5.164 | 0.317 | 20.83 | 5.481 | 0.1744 | 0.3109 | 0.2750 | -0.0156 |
| | Flux.1 | -0.3336 | 0.2601 | 4.744 | -2.040 | 20.49 | 7.446 | 0.1447 | 0.2091 | 0.2733 | -0.0085 |
| | Flux.1 (SFT) | -0.3623 | 0.2585 | 4.780 | -1.932 | 20.44 | 7.307 | 0.1516 | 0.2258 | 0.2747 | -0.0326 |
| | Flux.1 (DPO) | -0.2254 | 0.2666 | 4.961 | 1.013 | 20.65 | 7.387 | 0.1461 | 0.2081 | 0.2759 | -0.0089 |
| | Flux.1 (G=12 w/o Boundary) | -0.0451 | 0.2697 | 5.408 | 2.118 | 20.96 | 7.170 | 0.1474 | 0.2112 | 0.2810 | -0.0087 |
| | Flux.1 (G=12 w/o Dynamic) | 0.0466 | 0.2700 | 5.509 | 2.239 | 21.01 | 7.091 | 0.1479 | 0.2189 | 0.2800 | -0.0051 |
| | Flux.1 (G=12 w/o Cropping) | 0.0852 | 0.2704 | 5.513 | 2.655 | 21.09 | 7.012 | 0.1493 | 0.2192 | 0.2813 | -0.0041 |
| | Flux.1 (G=2 Ours) | -0.2379 | 0.2651 | 4.944 | 0.280 | 20.58 | 7.273 | 0.1441 | 0.2112 | 0.2750 | -0.0082 |
| | Flux.1 (G=4 Ours) | -0.0048 | 0.2678 | 5.287 | 0.534 | 20.67 | 7.281 | 0.1465 | 0.2145 | 0.2763 | -0.0074 |
| | Flux.1 (G=8 Ours) | 0.0389 | 0.2714 | 5.465 | 1.926 | 20.91 | 7.198 | 0.1480 | 0.2148 | 0.2789 | -0.0057 |
| | Flux.1 (G=12 Ours) | 0.0898 | 0.2718 | 5.503 | 2.779 | 21.11 | 6.970 | 0.1499 | 0.2205 | 0.2814 | -0.0042 |
| | Flux.1 (G=16 Ours) | 0.1462 | 0.2727 | 5.532 | 2.824 | 21.14 | 6.866 | 0.1512 | 0.2231 | 0.2816 | -0.0033 |
| **BrushBench** | BrushNetX | 0.5542 | 0.2655 | 5.824 | 1.099 | 20.95 | 5.440 | 0.0896 | 0.3385 | 0.2572 | -0.2222 |
| | PowerPaint v2 | 0.5942 | 0.2656 | 5.915 | 1.446 | 21.16 | 5.998 | 0.0853 | 0.2989 | 0.2626 | -0.1364 |
| | PrefPaint | 0.3075 | 0.2586 | 5.494 | -2.124 | 20.33 | 5.773 | 0.0863 | 0.3080 | 0.2539 | -0.1539 |
| | SD1.5 | 0.3840 | 0.2605 | 5.558 | -1.296 | 20.46 | 5.565 | 0.0880 | 0.3227 | 0.2563 | -0.1643 |
| | SD1.5 (SFT) | 0.3881 | 0.2608 | 5.546 | -1.165 | 20.51 | 5.567 | 0.0851 | 0.3796 | 0.2554 | -0.1864 |
| | SD1.5 (DPO) | 0.4463 | 0.2626 | 5.659 | 0.2575 | 20.69 | 5.546 | 0.0876 | 0.3225 | 0.2578 | -0.1632 |
| | SD1.5 (G=12 w/o Boundary) | 0.5202 | 0.2643 | 5.783 | 0.5152 | 20.86 | 5.533 | 0.0872 | 0.3206 | 0.2593 | -0.1619 |
| | SD1.5 (G=12 Ours) | 0.5386 | 0.2649 | 5.810 | 0.9615 | 20.97 | 5.456 | 0.0886 | 0.3252 | 0.2590 | -0.1157 |
| | Flux.1 | 0.4864 | 0.2615 | 5.629 | 0.3534 | 20.94 | 7.937 | 0.0714 | 0.1999 | 0.2525 | -0.1250 |
| | Flux.1 (SFT) | 0.4917 | 0.2616 | 5.617 | 0.0682 | 20.95 | 7.944 | 0.0708 | 0.1987 | 0.2527 | -0.1468 |
| | Flux.1 (DPO) | 0.5171 | 0.2625 | 5.718 | 0.6272 | 20.92 | 7.749 | 0.0710 | 0.2078 | 0.2544 | -0.1233 |
| | Flux.1 (G=12 w/o Boundary) | 0.5329 | 0.2651 | 5.761 | 0.8346 | 20.95 | 7.671 | 0.0730 | 0.2094 | 0.2579 | -0.1232 |
| | Flux.1 (G=12 w/o Dynamic) | 0.5471 | 0.2649 | 5.786 | 0.8150 | 20.93 | 7.541 | 0.0697 | 0.2163 | 0.2571 | -0.1169 |
| | Flux.1 (G=12 w/o Cropping) | 0.5524 | 0.2652 | 5.782 | 0.8627 | 20.95 | 7.932 | 0.0707 | 0.1971 | 0.2573 | -0.1123 |
| | Flux.1 (G=2 Ours) | 0.4933 | 0.2623 | 5.658 | 0.5807 | 20.93 | 8.076 | 0.0697 | 0.1919 | 0.2528 | -0.1201 |
| | Flux.1 (G=4 Ours) | 0.5072 | 0.2629 | 5.721 | 0.6015 | 20.93 | 7.757 | 0.0714 | 0.2040 | 0.2536 | -0.1162 |
| | Flux.1 (G=8 Ours) | 0.5288 | 0.2637 | 5.753 | 0.8063 | 20.94 | 7.806 | 0.0742 | 0.2085 | 0.2548 | -0.1151 |
| | Flux.1 (G=12 Ours) | 0.5503 | 0.2653 | 5.798 | 0.8712 | 20.98 | 7.853 | 0.0713 | 0.2022 | 0.2576 | -0.1114 |
| | Flux.1 (G=16 Ours) | 0.5617 | 0.2659 | 5.815 | 0.9158 | 21.06 | 7.723 | 0.0701 | 0.2095 | 0.2581 | -0.1064 |

