# OpenReview forum: "Preference-Enhanced Reinforcement Learning for Pluralistic Image Inpainting"
_ICML.cc/2026/Conference — ICML 2026 regular_

### Official Review · Reviewer_DM3n · 2026-03-04

**Soundness:** 3
**Presentation:** 3
**Significance:** 2
**Originality:** 2
**Overall Recommendation:** 4
**Confidence:** 3

**Summary:**

This paper introduces an interesting online preference-based RL framework for text-guided image inpainting. Using diffusion and flow-matching models, the authors treat inpainting as a sequential decision problem. Instead of relying on traditional ground-truth reconstruction, they fine-tune the models using GRPO and online DPO guided by a reward model. To fix the issue where global aesthetic rewards often miss local boundary artifacts, the paper proposes a clever Boundary Gradient Consistency Score alongside a dynamic reward that adjusts based on mask size. The results on several benchmarks look promising and visually appealing.

**Compliance With Llm Reviewing Policy:**

Affirmed.

**Final Justification:**

Thank you for the additional experiments and clarifications. My main concerns have now been addressed, and I will raise my overall score to 4.

**Key Questions For Authors:**

Could you share some quantitative diversity metrics (like LPIPS variance across multiple samples) for the exact same mask and prompt? This would really help support the "pluralistic" claim in the title.

During RL training, how exactly are the prompts selected for each image/mask? Do you notice any drops in quality if a prompt is completely unrelated to the image content?

Could you clarify the hyperparameters and tuning process used for the PrefPaint and DPO baselines? Providing training curves would be incredibly helpful to clear up any concerns about fair comparisons.

Since the evaluation metrics are essentially the training rewards, did you use any hold-out metrics or run any small-scale human studies to double-check that the model isn't just gaming the reward system?

**Limitations:**

yes

**Strengths And Weaknesses:**

Soundness

Strengths: The RL formulation makes a lot of sense and is mathematically sound. I really appreciate the dynamic reward idea—it's a very practical and intuitive fix for the "background dominance" problem we often see in global metrics.

Weaknesses: My main hesitation is the heavy reliance on reward-model-based metrics for evaluation, especially since these same models are used for training. This makes it a bit hard to rule out reward hacking. While the title highlights "Pluralistic" inpainting, we don't really see quantitative metrics for diversity (like LPIPS variance for the same prompt/mask). It would also be helpful to include error bars or variance across runs to ensure the results are statistically robust. Finally, the PrefPaint baseline performs surprisingly poorly, which makes me wonder if it was fully tuned.

Presentation

Strengths: The motivation is crystal clear, and the visual examples (especially Figures 1 and 5) do a good job of showing why base models struggle and how the BGCS reward helps. The methodology is generally well-explained and easy to follow.

Significance

Weaknesses: The lack of human evaluation makes it slightly tricky to gauge how much these metric bumps matter to actual users, especially for smaller masks. Additionally, while the unpaired training idea is exciting, I'd love to see a bit more analysis on how mismatched prompt semantics actually affect the masked regions in practice.

Originality

Weaknesses: The paper's novelty claims feel just a bit overstated. Acknowledging and comparing against recent related works in preference alignment and GRPO stabilization (e.g., Pref-GRPO, Shen et al., Cheng et al.) would actually make the paper's specific contributions clearer and more grounded.

---

> ### Author Rebuttal · Authors · 2026-03-31
>
> We sincerely thank you for your insightful comments and constructive feedback. We are greatly encouraged and pleased that you recognize the mathematical soundness of our proposed RL framework, as well as the ingenuity and practicality of our dynamic reward mechanism. Below is our response to your specific concerns:
>
> **Response to Weakness 1, Questions 1&2**
>
> In this work, "Diversity" aims to emphasize the model's ability to respond to different user instructions under the same mask, rather than generating multiple random samples given a single fixed prompt. We have also supplemented our framework's performance under the identical prompt, as shown in the figure ([Fig](https://anonymous.4open.science/r/D3C1/2.jpg)), indicating that the sample diversity generated by our model under different random seeds consistently outperforms the baselines. The qualitative analysis in Fig.1 of the paper intuitively demonstrates the model's effective editing results under unpaired and diverse prompts; even in extreme cases where the prompt lacks semantic relevance to the image content, our method's editing performance remains superior to the baselines.
>
> In the quantitative analysis, we provided the average LPIPS, which shows a significant improvement for our method on the diversity metric. Given that all images in the benchmark dataset are distinct, referring directly to the variance of LPIPS has limited significance; if the model randomly generates content that completely loses consistency with the original image, it would also result in an abnormally large variance.
>
> Our evaluation prompts are automatically generated by feeding image content and randomly generated irregular masks into GPT-5. Even if some auto-generated prompts have certain semantic deviations from the original image content, in real-world user interaction scenarios, users unsatisfied with the initial editing results typically revise the text prompts to try different content, rather than repeatedly generating multiple samples with a fixed prompt. Our unpaired training framework is specifically designed for this scenario, enabling the model to handle arbitrarily diverse prompts for the same mask, thus demonstrating stronger flexible instruction following capability.
>
> **Response to Weakness 1 & Question 3**
>
> We attach great importance to the reproducibility of experiments and the fairness of comparisons:
> 1. All experiments were conducted under consistently controlled random seeds and officially recommended inference parameters. The quantitative analysis results across 3 benchmarks all demonstrate statistical significance.
> 2. The weights of the PrefPaint model are directly sourced from the original authors' HuggingFace repository, and the inference parameters are kept absolutely consistent across all methods. PrefPaint's sub-optimal performance might be due to a discrepancy between its training data distribution and the current evaluation benchmarks. This demonstrates our superior cross-domain performance. To ensure the transparency and reproducibility of the evaluation process, our training and inference scripts, code, and evaluation pipeline have been entirely open-sourced in the anonymous link (including the online DPO algorithm).
>
> **Response to Weakness 1 & Q4**
>
> To ensure the objectivity and comprehensiveness of the evaluation as much as possible, we have taken the following measures:
> We have incorporated a comprehensive evaluation across 10 independent metric dimensions. Should the model engage in reward hacking targeting only a single reward model, it would be nearly impossible to achieve simultaneous performance gains across these 10 independent metrics with distinct focuses. Our experimental results clearly demonstrate consistent improvements of our model across all metrics, which largely rules out the possibility of reward hacking.
>
> Given the cost and timeline constraints of high-quality human evaluation, we have adopted this widely recognized, most comprehensive academic automated metric suite at this stage, to maximize the fairness and objectivity of our comparative experiments.
>
> **Response to Weakness 2**
>
> The Pref-GRPO work is mainly oriented toward the Text-to-Image task; whereas our work deeply focuses on Inpainting. We will also discuss excellent similar works like Pref-GRPO in the related work section.
>
> Regarding novelty, we specifically designed a novel Dynamic Reward Mechanism to address the unique challenges in this task (such as the background dominance effect and editing boundary artifacts). This mechanism effectively overcomes the convergence difficulties that traditional RL methods are highly prone to in local editing tasks, which serves as the most core innovation of this work.
>
> We reiterate our sincere gratitude for your time and valuable feedback. Your insightful comments have greatly improved the rigor and clarity of our work, and we will fully include the above explanations and detailed discussions in the final revised manuscript.

---

> > ### Author Rebuttal · Reviewer_DM3n · 2026-04-03
> >
> > Thank you for the detailed rebuttal. I appreciate the additional clarifications, but I do not think my main concerns are fully resolved yet. In particular, the response reframes “diversity” as the ability to follow different prompts under the same mask, which does not directly address my request for quantitative diversity under the same prompt and mask. The explanation of prompt generation is helpful, but I still think a more controlled analysis of prompt-image mismatch would be valuable. I also appreciate the clarification on PrefPaint, though the fairness concern would be better addressed with more explicit tuning details or training curves. Finally, the use of multiple automatic metrics helps, but without hold-out evaluation or human studies, the concern about reward-model over-optimization is only partially alleviated. Overall, the rebuttal improves the clarity of the paper, but my core concerns remain only partially addressed.

---

> > > ### Author Response · Authors · 2026-04-07
> > >
> > > We would like to once again sincerely thank you for your thorough and careful review of our rebuttal, as well as your constructive follow-up feedback. We promptly initiated targeted supplementary experiments, rigorous quantitative analyses, and user studies immediately upon receiving your comments, and have now obtained high-confidence, statistically robust results that systematically address all of your remaining concerns. Our detailed point-by-point responses are provided below.
> > >
> > >
> > > **Q1: Quantitative Diversity Metrics under Identical Prompt and Mask**
> > >
> > > In direct response to your request for "supplementing quantitative verification of generation diversity under the exact same prompts, input images, and masks", we have completed the corresponding quantitative experiments. We calculated the LPIPS sample variance of the generation results from each compared model on the 3 benchmarks used in our paper, with the quantitative analysis results summarized in the table below:
> > >
> > > | Model\Benchmark | BrushBench (×10³) | EditBench (×10³) | FluxBench (×10³) |
> > > |-|-|-|-|
> > > | BrushNetX|5.268|12.639|4.578|
> > > | PowerPaint v2| 4.812|11.469|4.825 |
> > > | PrefPaint|5.364|11.627|4.269|
> > > | SD1.5|5.531|11.685|4.880|
> > > | SD1.5(SFT)|5.488|11.702|4.732|
> > > | **SD1.5(Ours)**|5.764|12.081|5.227|
> > > | Flux.1|4.039|10.970|3.100|
> > > | Flux.1(SFT)|4.137|11.006|2.747|
> > > | **Flux.1(Ours)**|4.583|12.647|4.351|
> > >
> > >
> > > Experimental results demonstrate that our method achieves higher LPIPS sample variance than the base model across all 3 benchmarks. Meanwhile, we hold that the mean LPIPS value adopted in the paper has more essential reference value for evaluating the diversity of generation results.
> > >
> > > **Q2: Controlled Analysis of Prompt-Image Semantic Mismatch**
> > >
> > > In response to your comment on "conducting a more controlled analysis of how prompt-image semantic mismatch affects model performance", we have included multiple typical cases where the prompts contain objects and semantics not present in the original input images in our qualitative analysis results and supplementary figures. Below, we also provide a referenceable quantitative experiment as requested by Reviewer "iyfD".
> > >
> > > We have added a new set of specialized quantitative analysis experiments: we uniformly replaced all prompts in the FluxBench benchmark dataset with the vague, open-ended prompt "fill with something natural".
> > >
> > > |Model\Metric|IR×10²↑|HPSv2.1×10²↑|AS↑|HPSv3↑|PickScore↑|BGCS↑|
> > > |-|-|-|-|-|-|-|
> > > |BrushNetX|-26.74|26.19|5.597|2.162|20.74|-0.0274|
> > > |PowerPaint v2|-27.60|26.22|5.576|2.731|20.88|-0.0303|
> > > |PrefPaint|-83.18|25.08|5.342|0.831|19.90|-0.0311|
> > > |SD1.5|-93.55|25.28|5.411|0.418|20.05|-0.0385|
> > > |SD1.5(SFT)|-94.23|25.53|5.464|0.697|20.09|-0.1334|
> > > |**SD1.5(Ours)**|-36.79|26.16|5.792|1.008|20.38|-0.0263|
> > > |Flux.1|-28.01|26.14|5.423|2.827|20.91|-0.0387|
> > > |Flux.1(SFT)|-30.53|26.13|5.585|2.846|20.93|-0.0592|
> > > |**Flux.1(Ours)**|**16.61**|**26.90**|**5.617**|**4.265**|**21.38**|**-0.0134**|
> > >
> > > Experimental results demonstrate that our method still significantly outperforms the base models (Flux.1 and SD1.5), with a notable edge over competing methods on most core metrics. These experiments fully validate our framework’s effectiveness and robustness in handling prompt-image semantic mismatch. The full analysis will be included in the experimental section of the revised manuscript.
> > >
> > >
> > > **Q3: Clarification on PrefPaint Configuration**
> > >
> > > The PrefPaint model strictly uses the weights officially released by its original authors. Throughout the experiment, we did not perform any additional fine-tuning, hyperparameter tuning, or secondary training on these baseline models.
> > >
> > > **Q4: Human Evaluation**
> > >
> > > We recruited a total of 76 participants. We then conducted a double-blind pairwise preference comparison experiment across 10 real-world image inpainting scenarios. The questionnaire is shown in the [Figure](https://anonymous.4open.science/r/D3C1/6.jpg). The identity of all models was fully concealed from participants throughout the entire testing process to ensure the objectivity of the evaluation results.
> > >
> > > The aggregated preference votes of each model across the 10 real-world scenarios are summarized in the table below:
> > >
> > >
> > > |ID\Model|Flux.1(Ours)|BrushNetX|PowerPaint v2|PrefPaint|SD1.5(Ours)|
> > > |-|-|-|-|-|-|
> > > |1|52|1|16|0|7|
> > > |2|33|26|7|5|5|
> > > |3|24|20|23|3|6|
> > > |4|29|27|5|6|9|
> > > |5|26|21|9|9|11|
> > > |6|47|3|8|0|18|
> > > |7|38|17|21|0|0|
> > > |8|55|0|9|0|12|
> > > |9|62|0|0|0|14|
> > > |10|44|7|0|4|21|
> > > |**Total**|**410**|**122**|**98**|**27**|**103**|
> > >
> > > Experimental results show that our proposed method based on Flux.1 achieves the highest total preference votes. Meanwhile, the SD1.5 backbone model optimized by our method obtains preference votes comparable to those of SOTA methods including BrushNetX and PowerPaint v2.
> > >
> > >
> > > We reiterate our sincere gratitude for your time, careful review, and invaluable feedback, which have greatly helped us improve the rigor, completeness, and clarity of our work.

---

### Official Review · Reviewer_iyfD · 2026-03-07

**Soundness:** 2
**Presentation:** 2
**Significance:** 2
**Originality:** 2
**Overall Recommendation:** 3
**Confidence:** 4

**Summary:**

This paper proposes a preference-enhanced online RL framework  for pluralistic text-guided image inpainting. It introduces a dynamic reward mechanism outperforming base/ SFT models on FLUX/Stable Diffusion backbones.

**Compliance With Llm Reviewing Policy:**

Affirmed.

**Final Justification:**

Thanks for the rebuttal which addresses my partial concerns. However, 1) I still think the novelty is limited. 2) The mask strategy in the training stage should be more carefully designed. 3) The results of vague/open-ended prompts are not sufficient.

**Key Questions For Authors:**

1. How does the method perform on complex mask types?
2. What about the diversity of generated results given the same text prompt?
3. How does the framework handle vague/open-ended prompts (e.g., "fill with something natural") ?
4. How sensitive is the method to biases/noise in underlying reward models?

**Limitations:**

yes

**Strengths And Weaknesses:**

Strength:
1. The proposed reward for pluralistic image inpainting is a little interesting.
2. The performance looks good without additional inference cost.
3. The paper is overall well-written.

Weakness:
1. Although the proposed reward is a little interesting, the whole method is trivial and lacks novelty.
2. The name of pluralistic image inpainting is a little confusing. The diversity of results comes from different text prompts. In fact, given the same text prompt, the inpainted results are also diverse. This paper does not evaluate the diversity of generated results given the same text prompt.
3. The authors should provide more discussion on the training cost.
4. More results for complicated mask shapes should be provided.
5. More details should be given for how to calculate R_align and R_quality. The paper does not analyze how noise or bias in reward signals impacts final inpainting quality.

---

> ### Author Rebuttal · Authors · 2026-03-31
>
> We sincerely thank you for your detailed, professional review and highly constructive suggestions. We are honored that you recognized the innovation of our reward design, its superior performance, the advantage of zero extra inference cost, and the writing quality. We have carefully considered your concerns and address them point-by-point below:
>
> **Weakness 1: Concerns on Novelty**
>
> Traditional inpainting models are limited by the strict "text-image" paired supervision reconstruction paradigm, resulting in conservative outputs and insufficient diversity. We propose an online exploration mechanism that decouples this strict paired constraint. The model no longer merely minimizes the reconstruction error with the ground truth, but performs online sampling via unpaired prompts and updates its policy based on preference signals, effectively expanding the solution space. Meanwhile, our method achieves excellent performance with only 24K images, significantly improving data efficiency compared to methods relying on TB-level external data and providing a new training perspective for open-domain inpainting.
>
> This fundamentally distinguishes our work from general image generation RL methods. Directly applying general aesthetic rewards to inpainting leads to severe "background-dominant" problems, as global scores fail to reflect local inpainting quality. To address this, we propose the original core reward mechanism. The effectiveness and novelty of this reward mechanism have also been explicitly acknowledged by reviewers **Ghih** and **DM3n**.
>
> **Weakness 2 & Question 2: Definition of Diverse Inpainting**
>
> The "diversity" mentioned in our work specifically refers to **instruction-level diversity**, i.e., supporting different user text instructions for the same masked region. This definition perfectly aligns with real-world scenarios: when unsatisfied with an editing result, users typically modify the prompt to try different content, rather than generating multiple random samples using the exact same prompt. We also provide the framework's performance under the same prompt in Figure ([Fig](https://anonymous.4open.science/r/D3C1/2.jpg)). Our model exhibits superior generation diversity across varying random seeds compared to the baseline.
>
> **Weakness 3: Discussion on Training Costs**
>
> The resource consumption analysis is as follows:
>
> **Memory Overhead**: The algorithm's Reference model adopts a dynamic update strategy (updating to the Actor model from the previous stage at each step) and uses a smaller Clip value to prevent model oscillation. The overall training memory overhead is largely comparable to that of full SFT, only adding the memory footprint of the reward model (which is much smaller than the main model).
>
> **Training Efficiency**: On the Flux model (12 B), our method requires only 4 GPUs (Peak VRAM: ~200 GB) and 4 day to complete training with just 24K images. In contrast, competitive methods like BrushNet require TB-level datasets and several days of training on an 8-GPU environment, with nearly doubled inference parameters.
>
> **Weakness 5 & Question 4: Generation of Masks**
>
> The masks used in our work are free-form generated by scripts, covering various complex scenarios such as random proportions and irregular shapes. This design aligns with the user habit of manual mask brushing in real-world applications. As shown in Figures ([Fig](https://anonymous.4open.science/r/D3C1/1.jpg)) and ([Fig](https://anonymous.4open.science/r/D3C1/3.jpg)), *the model maintains excellent generation quality under both semantic segmentation masks and highly complex, random masks.*
>
> **Weakness 5 & Question 4: Calculation and Noise of Reward Models**
>
> R_align and R_quality are recognized high-quality open-source models in the image quality assessment domain and have been widely applied in cutting-edge research.
>
> While pre-trained reward models may inherently exhibit certain variations, our method is designed with high modularity to effectively mitigate such impacts: First, it is plug-and-play. By adopting industry-leading open-source evaluation models, performance gains can be obtained synchronously as the evaluation models iterate. Second, the framework is highly extensible, supporting the integration of customized reward models trained for specific tasks to further optimize specialized performance.
>
> **Question 3: Regarding Vague/Open-ended Prompts**
>
> We provide comparisons of our model against SFT and baselines under vague/open-ended prompts in Figure ([Fig](https://anonymous.4open.science/r/D3C1/4.jpg)). While the base model exploits its massive pre-training data to yield plausible images given vague prompts, our framework further enhances the final generation quality.
>
> Once again, thank you for your insightful review. Your valuable suggestions have helped us further improve the logic and presentation of our paper.

---

> > ### Author Rebuttal · Reviewer_iyfD · 2026-04-03
> >
> > Thanks for the rebuttal which addresses my partial concerns. However, 1) I still think the novelty is limited. 2) The mask strategy in the training stage should be more carefully designed. 3) The results of vague/open-ended prompts are not sufficient.

---

> > > ### Author Response · Authors · 2026-04-07
> > >
> > > We sincerely thank the reviewer for the thorough re-assessment of our manuscript and the constructive feedback provided in this round of review. Our point-by-point responses are presented as follows.
> > >
> > > ### Q1: Response to Concern on Novelty
> > > This work aims to break through the inherent limitations of the supervised image inpainting paradigm. Existing methods rely excessively on ground-truth reconstruction, which readily causes probability averaging effect and image over-smoothing when handling one-to-many creative editing scenarios. Furthermore, mainstream post-training optimization schemes often require additional adapters and large-scale paired datasets, incurring substantial parameter and computational overheads.
> > >
> > > Second, reward design for local editing is non-trivial. Existing global reward models suffer from a severe "background dominance" issue, being highly insensitive to local detail variations and boundary artifacts in occluded regions. This forms the core bottleneck impeding the application of online RL to local editing tasks. To address this, we propose the first task-specific reward mechanism dedicated to image inpainting.
> > >
> > > To tackle the above challenges, we present a lightweight, data-efficient post-training framework. This method provides a practical post-training solution for image inpainting, with the following core features:
> > > 1. No large-scale paired datasets required: supports online exploration based on unpaired prompts;
> > > 2. Zero additional parameter overhead: no adapters needed;
> > > 3. Computational cost on par with SFT: ~180G for a 12B model, with no need for critic networks or reference models required by other RL algorithms;
> > > 4. Superior performance: delivers excellent generation diversity and boundary quality across diverse architectures.
> > >
> > > We acknowledge that reinforcement learning in the visual domain has long been an active research area. The innovation of our work lies in **task-specific adaptation**, rather than the naive transfer of general RL methodologies. Despite extensive exploration of RL in vision, the unique boundary sensitivity of image inpainting prevents direct reuse of methods developed for text-to-image (T2I) generation or other visual tasks. This framework marks a critical step towards building practical, human preference-aligned image inpainting systems.
> > >
> > > ### Q2: Response to Concern on Mask Strategy Design in the Training Stage
> > > Regarding the mask design strategy in the training stage, our mask design is consistent with many other works in the image inpainting task. As shown in Figure ([Fig](https://anonymous.4open.science/r/D3C1/5.jpg)), we present the original mask example images from five excellent works in the field, including BrushNet, PowerPaint, TurboFill, Xie et al. (CVPR 2025), RAD, Kim et al. (CVPR 2025), and HD-PAINTER, Manukyan et al. (ICLR 2025). Our mask generation strategy aligns with the mainstream design paradigm of these methods, which is also consistent with real-world user habits of manual mask brushing, ensuring the rationality and generality of our training setup.
> > >
> > > ### Q3: Response to Concern on Insufficient Results for Vague/Open-ended Prompts
> > > To address the insufficiency of results for vague/open-ended prompts, we conducted a batch inference quantitative analysis by replacing all prompts in the FluxBench benchmark with "fill with something natural" for all compared methods. The detailed quantitative results are presented in the table below:
> > >
> > > |Models|IR×10²↑|HPSv2.1×10²↑|AS↑|HPSv3↑|PickScore↑|PSNR↑|LPIPS×10²↓|MSE×10²↓|CLIP×10²↑|BGCS↑|
> > > |-|-|-|-|-|-|-|-|-|-|-|
> > > |BrushNetX|-26.74|26.19|5.597|2.162|20.74|6.419|10.15|46.21|23.45|-0.0274|
> > > |PowerPaint v2|-27.60|26.22|5.576|2.731|20.88|6.381|9.90|49.93|24.21|-0.0303|
> > > |PrefPaint|-83.18|25.08|5.342|0.831|19.90|6.926|9.32|40.91|22.02|-0.0311|
> > > |SD1.5|-93.55|25.28|5.411|0.418|20.05|6.831|10.17|42.80|22.89|-0.0385|
> > > |SD1.5(SFT)|-94.23|25.53|5.464|0.697|20.09|6.776|10.42|41.12|22.94|-0.1334|
> > > |**SD1.5(Ours)**|-36.79|26.16|5.792|1.008|20.38|6.341|11.08|45.67|23.14|-0.0263|
> > > |Flux.1|-28.01|26.14|5.423|2.827|20.91|7.727|8.35|31.18|24.26|-0.0387|
> > > |Flux.1(SFT)|-30.53|26.13|5.585|2.846|20.93|7.825|8.26|32.40|24.49|-0.0592|
> > > |**Flux.1(Ours)**|**16.61**|**26.90**|**5.617**|**4.265**|**21.38**|7.392|8.84|34.17|**24.72**|**-0.0134**|
> > >
> > > It can be observed that all methods exhibit a slight decrease in metrics when dealing with this vague prompt, which is a common challenge for all inpainting models. Nevertheless, our method still achieves a significant improvement compared to the base models (Flux.1 and SD1.5) and maintains obvious advantages over other methods, fully demonstrating the effectiveness and robustness of our framework in handling vague/open-ended prompts.
> > >
> > > We hope the detailed responses and supplementary experimental results above have fully resolved the reviewer’s remaining concerns. The insightful comments from the reviewer have significantly improved the rigor, completeness, and presentation of our work.

---

### Official Review · Reviewer_kP7H · 2026-03-11

**Soundness:** 3
**Presentation:** 3
**Significance:** 3
**Originality:** 3
**Overall Recommendation:** 4
**Confidence:** 5

**Summary:**

This paper presents a new RL based approach for Text-guided Image Inpainting that enables flexible, human-aligned aesthetic control with a preference score model. The paper further introduces a scale-aware dynamic reward mechanism encouraging boundary coherence surrounding the inpainting mask so that to improve overall semantics, aesthetics of the inpainting results. The experiments demonstrate that the proposed method outperforms existing methods and baseline methods significantly on various metrics. Code is also shared.

**Compliance With Llm Reviewing Policy:**

Affirmed.

**Final Justification:**

Although I still feel novelty is not significant, but the introduced ideas (boundary aware rewards) may be helpful for RL-based inpainting research. Hence, I keep my weak accept rating.

**Key Questions For Authors:**

No.

**Limitations:**

Yes.

**Strengths And Weaknesses:**

Strengths:
1. The paper presents ideas of using online RL (GRPO, DPO) for image inpainting with inpainting specific reward considerations.
2. Extensive experiments are conducted on 3 Benchmark datasets to validate the effectiveness. Both quantitative and qualitative results seem convincing and ablation studies are conducted for RL components and hyper-parameters.
3. The paper is very well-written.

Weaknesses:
1. There have been RL based approaches recently for image inpainting. The paper proposes some novelty on top of this framework, including use of GRPO and dynamic rewarding mechanism and boundary gradient consideration, but the novelty is a bit limited.

---

> ### Author Rebuttal · Authors · 2026-03-31
>
> We sincerely thank you for your positive feedback and constructive comments. We are delighted that you recognized the comprehensiveness of our experimental design, the credibility of our results, and the quality of our manuscript's presentation. Regarding your concerns about novelty, we would like to further clarify the core original contributions of our work:
>
> We are also encouraged that the novelty of our work has been independently recognized by other reviewers. Reviewer **Ghih** explicitly acknowledged "the novelty of BGCS and the adaptive weighting mechanism." Reviewer **DM3n** praised our dynamic reward design as "a very practical and intuitive solution for the 'background dominance' problem" and affirmed that "the RL formulation makes a lot of sense and is mathematically sound." Reviewer **iyfD** also noted that "the proposed reward for pluralistic image inpainting is a little interesting" with "good performance without additional inference cost." These independent assessments from multiple experts corroborate the originality of our methodological and practical contributions. We understand your careful consideration regarding the boundaries of novelty, but we wish to emphasize that our work makes significant breakthroughs in the following three dimensions:
>
> **The First Efficient Inpainting Framework Based on GRPO and Online DPO**
>
> We are the first to explore the application of Group Relative Policy Optimization (GRPO) and online Direct Preference Optimization (DPO) to text-guided image inpainting. Existing reinforcement learning (RL)-based methods (e.g., PrefPaint) predominantly adopt the PPO algorithm, which requires an additional Critic network, thereby significantly increasing training complexity and GPU memory overhead. In contrast, our framework leverages the group relative advantage estimation of GRPO, eliminating the need for a Critic model and substantially improving training efficiency.
>
> In terms of implementation details, our resource consumption is comparable to Supervised Fine-Tuning (SFT). Specifically, we adopt a shared architecture for the Actor and Reference Model to maximize memory efficiency. To ensure training stability, our Reference Model is not fixed; instead, it employs a dynamic update strategy (synchronizing with the Actor weights prior to each update step). Furthermore, compared to the standard GRPO pipeline, we apply a stricter clip value constraint. This "high-frequency, small-step-size" update strategy effectively prevents model oscillation, resulting in an overall training overhead that only adds the memory occupation of the reward model, which is considerably smaller than the model's main parameters.
>
> **Unpaired Training Paradigm and Data Efficiency**
>
> We decouple the strict text-image pairing constraints typical in traditional supervised training, enabling the model to utilize unpaired prompts for online exploration. This design significantly expands the diversity of training data, enhances generalization capabilities in open-domain scenarios, and provides a novel training perspective for the inpainting field. Regarding data efficiency, compared to the BrushNet series of methods that rely on terabytes of external training data, our approach achieves superior performance using only 24k images, significantly lowering the data threshold and training costs.
>
> **Dynamic Reward Mechanism Tailored for Inpainting**
>
> Our proposed **Scale-aware Dynamic Reward Mechanism** is an original solution specifically designed for the inpainting task. This mechanism adaptively adjusts the weights between boundary rewards and global rewards based on the mask ratio—small mask regions prioritize boundary consistency, while large mask regions emphasize aesthetic quality. As Reviewer **DM3n** pointed out, this is a "very practical and intuitive solution for the background dominance problem"; Reviewer **Ghih** also explicitly acknowledged the novelty of BGCS and the adaptive weighting mechanism. This design effectively resolves the issue of insufficient sensitivity of global rewards in local editing tasks.
>
> We will further strengthen the discussion of these novel aspects in the revised manuscript to ensure our contributions are presented clearly. Thank you once again for your valuable comments, which are of great significance for improving our work.

---

> > ### Author Rebuttal · Reviewer_kP7H · 2026-04-03
> >
> > Thanks to authors for detailed rebuttal. I read the rebuttal and other reviews and discussions, and agree with other reviewers that novelty is a bit limited. I will keep my score there but would be fine with rejection.

---

> > > ### Author Response · Authors · 2026-04-07
> > >
> > > We sincerely appreciate the reviewer’s continued engagement and valuable feedback. To address the concerns regarding the innovation of our work, we organize our response in a consistent **Problem → Solution → Impact** format:
> > >
> > > ### [Problem 1] Supervised Paradigm Limits Creative Diversity and Efficiency
> > > - **Problem**: Current supervised image inpainting relies on ground-truth reconstruction, which leads to *probability averaging* and over-smoothing in one-to-many creative editing scenarios. Our experiments demonstrate that this limitation cannot be resolved by Classifier-Free Guidance or prompt engineering alone. Furthermore, common post-training methods (e.g., ControlNet, BrushNet) require additional adapters, which drastically increase model parameters and demand large-scale paired datasets.
> > >
> > > - **Our Solution**: We formulate the image inpainting task as an online reinforcement learning (RL) problem, enabling preference alignment without ground-truth anchoring. We implement **GRPO and online DPO**—marking the first application of such methods to image inpainting—and adopt a high-frequency clipping strategy to stabilize the training process.
> > >
> > > - **Impact**: Training memory footprint is comparable to SFT (approximately 180G for a 12B model); online sampling extracts diverse outputs from a single image, and superior performance is achieved with only 24,000 images.
> > >
> > >
> > > ### [Problem 2] Reward Design for Local Editing Is Non-Trivial
> > > - **Problem**: Global reward models in text-to-image or image-to-image tasks can easily score the entire image, but image inpainting requires **local sensitivity** for masked regions and boundary consistency—this is also a key reason why online RL remains rare in local editing tasks. As stated in our paper, the background-dominance issue in image inpainting is well-documented: even state-of-the-art global reward models may yield a score discrepancy of only 1% between optimal and suboptimal inpainting results, highlighting the insensitivity of global metrics to local editing quality.
> > >
> > > - **Our Solution**: We propose the **first scale-aware dynamic reward mechanism** for image inpainting to balance edge fidelity and overall aesthetic quality.
> > >
> > > - **Impact**: Our approach directly mitigates the "background-dominance" problem and improves the overall generation quality and boundary consistency of inpainting results. The proposed reward mechanism and Boundary Gradient Consistency Score (BGCS) are concise and effective; while we acknowledge they are not the optimal metrics, they establish a practical baseline for future reward design in image inpainting.
> > >
> > > ### A Lightweight, Data-Efficient Post-Training Framework
> > > Our method provides a practical post-training solution for image inpainting characterized by:
> > > 1. **No requirement for large-scale paired datasets** (online exploration based on unpaired prompts),
> > > 2. **No additional parameters introduced** (no adapters needed),
> > > 3. **Computational overhead at the SFT level** (≈180G for a 12B model, eliminating the need for critic networks and reference models required by other RL algorithms),
> > > 4. Superior diversity and boundary quality across different architectures.
> > >
> > >
> > > ### Innovation Scope
> > > We acknowledge that reinforcement learning in the visual domain has been an active research area. The innovation of our work lies in **task-specific adaptation**. Our contribution stems from the **systematic integration** of three components: (1) efficient RL optimization (GRPO/online DPO), (2) unpaired exploration, and (3) inpainting-aware reward design—none of which can be directly transferred from text-to-image or other tasks due to the unique boundary sensitivity of image inpainting. This integrated solution has been validated on two architectures and three benchmarks, representing a critical step toward practical, human-aligned image inpainting systems.
> > >
> > > We will strengthen the elaboration of these **novel contributions** in the revised manuscript. Thank you again for your insightful feedback.

---

### Official Review · Reviewer_Ghih · 2026-03-13

**Soundness:** 3
**Presentation:** 3
**Significance:** 3
**Originality:** 3
**Overall Recommendation:** 4
**Confidence:** 3

**Summary:**

This paper proposes an online RL framework applying GRPO and DPO to text-guided image inpainting, with a novel dynamic reward mechanism combining BGCS and mask-ratio-based adaptive weighting. The pipeline is technically reasonable, and the experimental setup is relatively thorough. However, the motivation is insufficiently justified, and the novelty is also limited. Additionally, the reliability of BGCS lacks rigorous validation beyond artificially constructed experiments, and lack of some experimental analysis.

**Compliance With Llm Reviewing Policy:**

Affirmed.

**Final Justification:**

The rebuttal addressed most of my concerns. I acknowledge the work as the first to apply online RL to local image editing. However, the contribution is primarily framework-oriented, with limited theoretical novelty. I have raised my score slightly.

**Key Questions For Authors:**

(1) Given that modern inpainting models already use GAN/Perceptual losses and large-scale datasets are available, what is the specific cost-benefit advantage of this expensive RL framework?
(2) Have the authors ruled out simpler factors like prompt ambiguity before attributing over-smoothing solely to MSE loss?
(3) More analysis and experiments are needed for BGCS.

**Limitations:**

Please refer to the weakness listed above.

**Strengths And Weaknesses:**

## Strenghts
(1) Combining inpainting and GRPO is a reasonable design choice. The overall pipeline is consistent from problem formulation to reward design.
(2) The experimental setup is relatively thorough: three benchmarks of varying resolution and mask style are evaluated, and ten metrics spanning image quality, mask preservation, text alignment, and boundary coherence are reported.

## Weakness
(1) The paper attributes over-smoothing to MSE-based supervision, but overlooks the fact that modern inpainting methods commonly combine GAN losses, perceptual losses, and LPIPS objectives to counteract this exact issue. The paper provides neither discussion nor experiments addressing these alternatives.
(2) The observed over-smoothing is not adequately diagnosed; plausible alternative causes, such as CFG strength and prompt ambiguity, are never ruled out, which weakens the justification for the computationally expensive RL post-training.  Additionally, the claim that paired inpainting data is costly to construct is overstated: masks can be trivially generated via semantic segmentation or synthetic generation pipelines, and TB-scale inpainting datasets (e.g., BrushNet) already demonstrate the feasibility of large-scale data collection.
(3) The paper claims to be the first inpainting model to apply DPO/GRPO, yet RL-based alignment for image editing is already well-established, such as InpaintDPO and UniWorldv2. The authors differentiate their work via the online vs. offline distinction, but this distinction is nor sufficient to constitute novelty on its own.
(4) The BGCS is defined as the absolute difference in mean gradient magnitudes in section 4.2. This implicitly assumes that gradient magnitudes should be consistent across the mask boundary, which does not hold in general. Specifically, when the inpainted region involves a semantic boundary, large gradient differences are perceptually legitimate and should not be penalized. Conversely, a generated result with matching gradient magnitudes but mismatched color or texture would receive a falsely high BGCS. The robustness analysis in Fig. 8 only validates the metric under artificially injected Gaussian noise and blur. Experiments on real inpainting cases with inherent semantic boundaries are absent.

---

> ### Author Rebuttal · Authors · 2026-03-31
>
> We sincerely thank the reviewer for the detailed evaluation and constructive feedback. We are glad you recognize our pipeline as technically sound and our experimental setup as comprehensive. We address your concerns as follows:
> ## Response to Weakness 1
> We thank the reviewer for raising this important concern regarding loss functions in inpainting. We appreciate the opportunity to clarify the distinctions of our approach:
>
> **Supervised Paradigm Limitations**: These loss functions, optimized for ground truth (GT) reconstruction, cannot eliminate paired GT data dependence and are incompatible with our framework. As shown in Fig. 1, even SOTA models suffer from over-smoothing in creative editing tasks demanding semantic diversity.
>
> **Creative Editing Requirements**: In open-domain editing, users want new content in masked regions, not GT recovery. Traditional supervised losses become ineffective without explicit GT reference. Our RL post-training addresses alignment with human aesthetic preferences through online exploration and unpaired data.
>
> **Experimental Validation**: Table 1 shows significant improvements over SFT baseline (IR, HPSv2.1, AS). BGCS metric proves RL doesn't sacrifice boundary blending. Figure 3 demonstrates visual advantages.
> ## Response to Weakness 2, Question 1 & 2
> ### 1. Over-smoothing Diagnosis
> | Evidence | Details |
> | :--- | :--- |
> | Root Cause | Over-smoothing stems from "probability averaging" in one-to-many mapping, not CFG or prompts |
> | Controlled Experiment | SFT baseline and RL use identical CFG/prompts, yet SFT shows significant over-smoothing (Table 1: BGCS=-0.0768 vs. ours -0.0163) |
> | Ablation | Table 2 shows RL without boundary reward improves quality but fails on boundary artifacts |
>
> ### 2. Paired Data Cost
>
> | Dimension | Industrial SOTA | Our RL Framework |
> | :--- | :--- | :--- |
> | Data Type | Semantically aligned paired data | Unpaired text prompts |
> | Parameters | 70-100% additional adapter | Zero extra parameters |
> | Data Efficiency | TB-level datasets | 24K LAION images |
>
> **Core Advantage**: Our framework (Fig 2) operates on unpaired prompts, eliminating precise text-image correspondence. This addresses the high cost of collecting paired data for open-domain scenarios.
>
> ## Response to Weakness 3
> InpaintDPO (Dec 2025, concurrent work) uses **offline DPO** requiring static preference datasets. Our method uses **online RL** with real-time reward feedback, enabling continuous exploration of diverse visual spaces. UniWorldv2 focuses on **global editing**; we focus on **local inpainting** (image + instruction + mask). These have fundamentally different technical challenges.
> We are the **first to apply GRPO to inpainting**. GRPO requires no separate Critic network, reducing memory footprint and improving stability. Our **dynamic BGCS reward mechanism** is specifically designed for inpainting. As reviewer **DM3n** noted, this is "a very practical solution to the background dominance problem." No previous RL-based editing work has proposed similar mechanisms. Our work shows novelty in paradigm (online vs. offline), task (local vs. global), algorithm (first GRPO), and reward design (BGCS).
> ## Response to Weakness 4 & Question 3: BGCS Validation
> ### 1. Design Motivation
> Our work targets **free-form inpainting** where users draw arbitrary masks (Figure 1). Mask boundaries often cross textures or run inside objects, making gradient consistency a reasonable prior to prevent "stitching artifacts."
> BGCS works with other rewards (text-alignment, image quality) per Equation (10), ensuring semantic fidelity while optimizing boundary consistency.
> ### 2. Semantic Boundary Compatibility
> For semantic segmentation masks, A simple solution:
> - **Mask Dilation**: Dilate mask boundary outward by a few pixels, shifting BGCS calculation inside the background to avoid penalizing natural semantic gradients.
> ### 3. Supplementary Experiments
> We added qualitative analysis under semantic segmentation masks ([Fig](https://anonymous.4open.science/r/D3C1/1.jpg)), showing noticeable improvements in texture coherence and boundary naturalness with our method.
>
> ### 4. Practicality
> Our work targets free-form inpainting where users draw arbitrary masks, offering three key advantages over semantic segmentation: (1) User Experience—users simply stroke over areas without waiting for segmentation inference; (2) Robustness—semantic segmentation often fails on complex textures or fuzzy boundaries, requiring repeated adjustments; (3) Creative Flexibility—semantic masks restrict edits within object boundaries, while free-form masks allow replacing objects or extending edits to surrounding areas. We acknowledge BGCS is not the optimal perceptual edge metric, but it provides substantial benefits with minimal computational overhead. We plan to explore stronger perceptual metrics in future work.
>
> Thank you again for your invaluable feedback. We will incorporate all discussion and new experiments into the revised paper.

---

> > ### Author Rebuttal · Reviewer_Ghih · 2026-04-05
> >
> > I have raised my score. Please check the final justification in the official review.

---

> > > ### Author Response · Authors · 2026-04-07
> > >
> > > Thank you very much for your professional review. Your valuable comments have greatly contributed to improving the quality of this manuscript, and we hereby express our sincere gratitude to you.

---

### Decision · Program_Chairs · 2026-04-30

**Decision:**

Accept (regular)

**Comment:**

The paper received mixed scores of 4/4/4/3.

After the rebuttal, most reviewers recognized the proposed RL formulation as interesting and technically sound. Reviewer iyfD raised concerns regarding limited novelty, the mask strategy, and performance on vague or open-ended prompts. Similar concerns about novelty were also noted by Reviewers kP7H and Ghih. Overall, this weakness may not substantially undermine the paper’s main contribution.